# Multi-label Node Classification On Graph-Structured Data

**Tianqi Zhao** *T.Zhao-1@tudelft.nl*
*Department of Intelligent Systems*
*Delft University of Technology*

**Ngan Thi Dong** *dong@l3s.de*
*L3S Research Center, Hannover, Germany*

**Alan Hanjalic** *A.Hanjalic@tudelft.nl*
*Delft University of Technology*

**Megha Khosla** *M.Khosla@tudelft.nl*
*Delft University of Technology*

**Reviewed on OpenReview:** *https://openreview.net/forum?id=EZhkV2BjDP*

## Abstract

Graph Neural Networks (GNNs) have shown state-of-the-art improvements in node classification tasks on graphs. While these improvements have been largely demonstrated in a multi-class classification scenario, a more general and realistic scenario in which each node could have multiple labels has so far received little attention. The first challenge in conducting focused studies on multi-label node classification is the limited number of publicly available multi-label graph datasets. Therefore, as our first contribution, we collect and release three real-world biological datasets and develop a multi-label graph generator to generate datasets with tunable properties. While high label similarity (high homophily) is usually attributed to the success of GNNs, we argue that a multi-label scenario does not follow the usual semantics of homophily and heterophily so far defined for a multi-class scenario. As our second contribution, we define homophily and Cross-Class Neighborhood Similarity for the multi-label scenario and provide a thorough analyses of the collected 9 multi-label datasets. Finally, we perform a large-scale comparative study with 8 methods and 9 datasets and analyse the performances of the methods to assess the progress made by current state of the art in the multi-label node classification scenario. We release our benchmark at `https://github.com/Tianqi-py/MLGNC`.

## 1 Introduction

Most of the existing works on graph node classification deploying Graph Neural Networks (GNNs) focus on a multi-class classification scenario while ignoring a more general and realistic scenario of multi-label classification, in which each node could have multiple labels. This scenario holds, for example in protein-protein interaction networks, in which each protein is labeled with multiple protein functions or associated with different diseases, or in social networks, where each user may carry multiple interest labels. In this work, we focus on multi-label node classification on graph-structured data with graph neural networks. For the sake of brevity, in what follows we will refer to multi-label node classification on graphs simply as multi-label node classification.

Regarding the approaches to deploying GNNs in a multi-label node classification scenario, a common practice is to transform the classification problem into multiple binary classification problems, one per label (Zhang

& Zhou, 2013). In other words, $|L|$ binary classifiers are trained, where each classifier $j$ is responsible for predicting the 0/1 association for the corresponding label $\ell_j \in L$. An assumption here, however, is that given the learned feature representations of the nodes, the labels are conditionally independent (Ma et al., 2020). The validity of this assumption cannot be assured in a GNN-based learning approach as GNNs ignore the label correlation among the neighboring nodes and only focus on node feature aggregation during the representation learning step (Jia & Benson, 2020; Ma et al., 2020).

Furthermore, we note that the success of GNNs is widely attributed to feature smoothing over neighborhoods and high label similarity among the neighboring nodes. Graphs with high similarity among labels of neighboring nodes are referred to as having high homophily. Alternatively, in heterophilic graphs, the labels of neighboring nodes usually disagree. Consequently, approaches like H2Gcn (Zhu et al., 2020) have been proposed, which claim a high performance on both homophilic and heterophilic node classification datasets. A subtle point here, however, is that a network with nodes characterized by multiple labels does not obey the crisp separation of homophilic and heterophilic characteristics. As an illustration, consider a friendship network with node labels representing user interests. Each user might share only a very small fraction of the interests with his friends, which indicates low homophily in the local neighborhood. Yet her/his interests/labels could be fully determined by looking at his one-hop neighbors. Therefore, the network is also not heterophilic in the sense that the connected users have similar interests. Consequently, the solutions taking into account higher-order neighborhoods to tackle low homophily might not always perform well in multi-label networks.

The lack of focused studies on multi-label node classification can also be attributed to the scarcity of available benchmark multi-label graph datasets. As an example, there is a single multi-label classification dataset, namely OGB-Proteins in the Open Graph Benchmark (OGB) (Hu et al., 2020). More so, the OGB-Proteins dataset has around 90% of the nodes unlabeled in the test set. While the OGB leaderboard reflects benchmarking of a large number of methods on OGB-Proteins, the lack of labels in the test set combined with the use of the Area Under the ROC Curve (AUROC) metric leads to overly exaggerated performance scores. In particular, the performance is measured by the average of AUROC scores across the $L$ ($L$ is the total number of labels) binary classification tasks. In such a scenario, the model already achieves a very high score if it outputs a high probability for the negative class for each binary classification task.

**Our Contributions.** Our work thoroughly investigates the problem of multi-label node classification with GNNs. **Firstly**, we analyze various characteristics of multi-label graph-structured datasets, including label distribution, label induced first- and second-order similarities, that influence the performance of the prediction models. We observe that a large number of nodes in the current datasets only have a single label even if the average number of labels per node is relatively high. Moreover, for the popular OGB-Proteins dataset, around 89.38% of the nodes in the test set and 29.12% of train nodes have no label assigned.

**Secondly**, to remedy the gap of lack of datasets we build a benchmark of multi-label graph-structured datasets with varying structural properties and label-induced similarities. In particular, we curate 3 biological graph datasets using publicly available data. Besides, we develop a synthetic multi-label graph generator with tunable properties. The possibility to tune certain characteristics allows us to compare various learning methods rigorously.

**Finally**, we perform a large-scale experimental study evaluating 8 methods from various categories for the node classification task over 9 datasets. We observe that simple baselines like DeepWalk outperform more sophisticated GNNs for several datasets. We present a comprehensive analysis of the performance of different methods based on their own and the dataset's characteristics.

## 2 Background and Related Work

### 2.1 Notations and The Problem Setting.

**Notations.** Let $\mathcal{G} = (\mathcal{V}, \mathcal{E})$ denote a graph where $\mathcal{V} = \{v_1, \cdots, v_n\}$ is the set of vertices, $\mathcal{E}$ represents the set of links/edges among the vertices. We further denote the adjacency matrix of the graph by $\mathbf{A} \in \{0,1\}^{n\times n}$ and $a_{i,j}$ denotes whether there is an edge between $v_i$ and $v_j$. $\mathcal{N}(v)$ represents the immediate neighbors of node $v$

in the graph. Furthermore, let $\mathbf{X} = \{\mathbf{x}_1, \cdots, \mathbf{x}_n\} \in \mathbb{R}^{n \times D}$ and $\mathbf{Y} = \{\mathbf{y}_1, \cdots, \mathbf{y}_n\} \in \{0,1\}^{n \times C}$ represent the feature and label matrices corresponding to the nodes in $\mathcal{V}$. In the feature matrix and label matrix, the $i$-th row represents the feature/label vector of node $i$. Let $\ell(i)$ denote the set of labels that are assigned to node $i$. Finally, let $\mathcal{F}$ correspond to the feature set and $\mathcal{L}$ be the set of all labels.

**Problem Setting.** In this work, we focus on multi-label node classification problem on graph-structured data. In particular, we are given a set of labeled and unlabelled nodes such that each node can have more than one label. We are then interested in predicting labels of unlabelled nodes. We assume that the training nodes are completely labeled. We deal with the transductive setting multi-label node classification problem, where the features and graph structure of the test nodes are present during training.

### 2.2 Related Work

Multi-label classification, which assigns multiple labels for each instance simultaneously, finds applications in multiple domains ranging from text classification to protein function prediction. In this work we focus on the case when the input data is graph-structured, for example, a protein-protein interaction network or a social network. For completeness, we also discuss the related works on using graph neural networks for multi-label classification on non-graph-structured data in Section 2.2.2. Several other paradigms of multi-label classification including extreme multi-label classification (Liu et al., 2017; Song et al., 2022; Wang et al., 2019), partial multi-label classification (Huynh & Elhamifar, 2020; Jain et al., 2017), multi-label classification with weak supervision (Chu et al., 2018; Hu et al., 2019) ( for a complete overview see the recent survey (Liu et al., 2021)). are out of the scope of the current work.

#### 2.2.1 Multi-label Classification On Graph-structured Data

Recent methods designed for multi-label node classification over graph-structured data can be categorized into four groups utilizing (1) node embedding approaches, (2) convolutional neural networks, (3) graph neural networks, and (4) the combination of label propagation and graph neural networks.

**Node representation or embedding approaches** (Perozzi et al., 2014; Khosla et al., 2019; Ou et al., 2016) usually generate a lookup table of representations such that similar nodes are embedded closer. The learned representations are used as input features for various downstream prediction modules. While different notions of similarities are explored by different approaches, a prominent class of method is random walk based which defines similarity among nodes by their co-occurrence frequency in random walks. In this work, we specifically use DeepWalk (Perozzi et al., 2014) as a simple baseline that uses uniform random walks to define node similarity.

Other methods like (Shi et al., 2019; Zhou et al., 2021; Song et al., 2021) use **convolutional neural networks** to first extract node representations by aggregating feature information from its local neighborhood. The extracted feature vectors are then fused with label embeddings to generate final node embeddings. Finally, these node embeddings are used as input for the classification model to generate node labels. In this work, we adopt LANC (Zhou et al., 2021) as a baseline from this category, as previous works (Zhou et al., 2021; Song et al., 2021) have shown its superior performance compared to other commonly used baselines for the multi-label node classification task.

**Graph neural networks(GNNs)** popularised by graph convolution network (Kipf & Welling, 2016) and its variants compute node representation by recursive aggregation and transformation of feature representations of its neighbors which are then passed to a classification module. Let $\mathbf{x}_i^{(k)}$ be the feature representation of node $i$ at layer $k$, $\mathcal{N}(i)$ denote the set of its 1-hop neighbors. The $k-th$ layer of a graph convolutional operation can then be described as

$$\mathbf{z}_i^{(k)} = \text{AGGREGATE}\left(\left\{\mathbf{x}_i^{(k-1)}, \left\{\mathbf{x}_j^{(k-1)} \mid j \in \mathcal{N}(i)\right\}\right\}\right), \quad \mathbf{x}_i^{(k)} = \text{TRANSFORM}\left(\mathbf{z}_i^{(k)}\right)$$

For the multi-label node classification, a sigmoid layer is employed as the last layer to predict the class probabilities: $\boldsymbol{y} \leftarrow (\text{sigmoid}(\boldsymbol{z}_i^{(L)}\theta))$, where $\theta$ corresponds to the learnable weight matrix in the classification module. GNN models mainly differ in the implementation of the aggregation layer. The simplest model is

the graph convolution network (GCN) (Kipf & Welling, 2016) which employs degree-weighted aggregation over neighborhood features. GAT (Veličković et al., 2018) employs several stacked Graph Attention Layers, which allows nodes to attend over their neighborhoods' features. GRAPHSAGE (Hamilton et al., 2017) follows a sample and aggregate approach in which only a random sample of the neighborhood is used for the feature aggregation step. GNNs, in general, show better performance on high homophilic graphs in which the connected nodes tend to share the same labels. Recent approaches like H2GCN (Zhu et al., 2020) show improvement on heterophilic graphs (in the multi-class setting). Specifically, it separates the information aggregated from the neighborhood from that of the ego node. Further, it utilizes higher-order neighborhood information to learn informative node representations.

**Label propagation with GNNs.** Prior to the advent of GNNs label propagation (LPA) algorithms constituted popular approaches for the task of node classification. Both LPA and GNNs are based on message passing. While GNNs propagate and transform node features, LPA propagates node label information along the edges of the graph to predict the label distribution of the unlabelled nodes. A few recent works (Yang et al., 2021; Wang & Leskovec, 2020) have explored the possibilities of combining LPA and GNNs. (Yang et al., 2021) employs knowledge distillation in which the trained GNN model (teacher model) is distilled into a combination of GNN and parameterized label propagation module. GCN-LPA (Wang & Leskovec, 2020) utilizes LPA serves as regularization to assist the GCN in learning proper edge weights that lead to improved classification performance. Different from our work both of the above works implicitly assume a multi-class setting. Moreover, (Yang et al., 2021) focuses on the interpretable extraction of knowledge from trained GNN and can only discover the patterns learned by the teacher GNN.

#### 2.2.2 Multi-label Classification On Non-Graph-Structured Data Using GNNs

There has also been an increasing trend to use graph neural networks to exploit the implicit relations between the data and labels in multi-label classification problems on non-graph data. For example, (Saini et al., 2021) models the problem of extreme classification as that of link prediction in a document-label bipartite graph and uses graph neural networks together with the attention mechanism to learn superior node representations by performing graph convolutions over neighborhoods of various orders. ML-GCN(Shi et al., 2019) generates representations for the images using CNN and extracts label correlation by constructing a label-label graph from the label co-occurrence matrix. LaMP(Lanchantin et al., 2019) treats labels as nodes on a label-interaction graph and computes the hidden representation of each label node conditioned on the input using attention-based neural message passing. Likewise, for the task of multi-label image recognition (Chen et al., 2019) builds a directed graph over the object labels, where each label node is represented by word embeddings of the label, and GCN is employed to map this label graph into a set of inter-dependent object classifiers. The above works and many more like (Li et al., 2021; Zong & Sun, 2020; Ma et al., 2021a; Zheng et al., 2022; Shi et al., 2020; Xu et al., 2021; Cheng et al., 2021; Pal et al., 2020) are not part of the current study as they are developed for non-graph structured data.

## 3 A detailed analysis of existing and new datasets

We commence by analyzing various properties of existing multi-label datasets including label distributions, label similarities, and cross-class neighborhood similarity(CCNS), which could affect the performance of prediction models. In section 3.2 we further curate new real-world biological datasets which to some extent improve the representativeness of multi-label graph datasets. Finally, in Section 4 we propose our synthetic multi-label graph generator to generate multi-label graph datasets with tunable properties. The possibility to control specific influencing properties of the dataset allows us to benchmark various learning methods effectively. We will need the following quantification of label similarity in multi-label datasets.

**Label homophily.** The performance of GNNs is usually argued in terms of label homophily which quantifies similarity among the neighboring nodes in the graph. In particular, label homophily is defined in (Zhu et al., 2020) as the fraction of the homophilic edges in the graph, where an edge is considered homophilic, if it connects two nodes with the same label. This definition can not be directly used in the multi-label graph datasets, as each node can have more than one label and it is rare in the multi-label datasets that the whole

label sets of two connected nodes are the same. Usually, two nodes share a part of their labels. We, therefore, propose a new metric to measure the label homophily $h$ of the multi-label graph datasets as follows.

**Definition 1** *Given a multi-label graph $\mathcal{G}$, the label homophily $h$ of $\mathcal{G}$ is defined as the average of the Jaccard similarity of the label set of all connected nodes in the graph:*

$$h = \frac{1}{|\mathcal{E}|} \sum_{(i,j)\in\mathcal{E}} \frac{|\ell(i) \cap \ell(j)|}{|\ell(i) \cup \ell(j)|}.$$

Label homophily is a first-order label-induced similarity in that it quantifies the similarity among neighboring nodes based on their label distributions.

**Cross-Class Neighborhood Similarity for Multi-label graphs.** Going beyond the label similarity among neighboring nodes, we consider a second order label induced metric which quantifies the similarity among neighborhoods of any two nodes. Ma et al. (2021b) introduced Cross-Class Neighborhood Similarity (CCNS) for multi-class graphs. Using CCNS, (Ma et al., 2021b) attributes the improved performance of GNNs to the higher similarity among neighborhood label distributions of nodes of the same class as compared to different classes. We extend their proposed CCNS measure to analyze multi-label datasets. Given two classes $c$ and $c'$, the CCNS score measures the similarity in the label distributions of neighborhoods of nodes belonging to $c$ and $c'$ and is defined as follows. One can visualize the CCNS scores between all class pairs in an $C \times C$ matrix where $C$ is the total number of classes. Common GNNs are expected to perform better on datasets which higher scores on the diagonal than off-diagonal elements of the corresponding CCNS matrix.

**Definition 2** *Given a multi-label graph $\mathcal{G}$ and the set of node labels $Y$ for all nodes, we define the multi-label cross-class neighborhood similarity between classes $c, c' \in C$ is given by*

$$s(c, c') = \frac{1}{|\mathcal{V}_c||\mathcal{V}_{c'}|} \sum_{i\in\mathcal{V}_c, j\in\mathcal{V}_{c'}, i\neq j} \frac{1}{|\ell(i)||\ell(j)|} \cos(\mathbf{d}_i, \mathbf{d}_j), \quad (1)$$

*where $\mathcal{V}_c = \{i | c \in \ell(i)\}$ is the set of nodes with one of their labels as c. The vector $\mathbf{d}_i \in \mathbb{R}^C$ corresponds to the empirical histogram (over $|C|$ classes) of node i's neighbors' labels, i.e., the $c^{th}$ entry of $\mathbf{d}_i$ corresponds to the number of nodes in $\mathcal{N}(i)$ that has one of their label as c and the function cos(.,.) measures the cosine similarity and is defined as*

$$cos(\mathbf{d}_i, \mathbf{d}_j) = \frac{\mathbf{d}_i \cdot \mathbf{d}_i}{||\mathbf{d}_i||||\mathbf{d}_j||}.$$

Note that as a node in a multi-label dataset could belong to multiple classes we exclude the possibility of comparing a node to itself. By introducing a factor of $|\ell(i)||\ell(j)|$ in the denominator we are able to normalize the contribution of multi-labeled nodes for several class pairs.

### 3.1 Existing datasets

We start by analyzing the four popular multi-label node classification datasets: (i) BlogCat (Tang & Liu, 2009), in which nodes represent bloggers and edges their relationships, the labels denote the social groups a blogger is a part of, (ii)Yelp (Zeng et al., 2019), in which nodes correspond to the customer reviews and edges to their friendships with node labels representing the types of businesses and (iii) OGB-Proteins (Hu et al., 2020), in which nodes represent proteins, and edges indicate different types of biologically meaningful associations between the proteins, such as physical interactions, co-expression, or homology (Consortium, 2018; D et al., 2019). The labels correspond to protein functions. (iiii) DBLP (Akujuobi et al., 2019), in which nodes represent authors and edges the co-authorship between the authors, and the labels indicate the research areas of the authors.

We report in Table 1 the characteristics of these datasets including the label homophily. In the following, we discuss in detail the various characteristics of these datasets along with their limitations for the effective evaluation of multi-label classification.

Table 1: Dataset statistics. $|\mathcal{V}|$ and $|\mathcal{E}|$ denote the number of nodes and edges in the graph. $|\mathcal{F}|$ is the dimension of the node features. *clus* and $r_{homo}$ denote the clustering coefficient and the label homophily. $C$ indicates the size of all labels in the graph. $\ell_{med}$, $\ell_{mean}$, and $\ell_{max}$ specify the median, mean, and max values corresponding to the number of labels of a node. '25%', '50%', and '75%' corresponds to the 25th, 50th, and 75th percentiles of the sorted list of the number of labels for a node. "N.A." means the corresponding characteristic is not available in the graph.

| Dataset | $\lvert\mathcal{V}\rvert$ | $\lvert\mathcal{E}\rvert$ | $\lvert\mathcal{F}\rvert$ | *clus* | $r_{homo}$ | $C$ | $\ell_{med}$ | $\ell_{mean}$ | $\ell_{max}$ | 25% | 50% | 75% |
|---|---|---|---|---|---|---|---|---|---|---|---|---|
| BlogCat | 10K | 333K | N.A. | 0.46 | 0.10 | 39 | 1 | 1.40 | 11 | 1 | 1 | 2 |
| Yelp | 716K | 7.34M | 300 | 0.09 | 0.22 | 100 | 6 | 9.44 | 97 | 3 | 6 | 11 |
| OGB-Proteins | 132K | 39M | 8 | 0.28 | 0.15 | 112 | 5 | 12.75 | 100 | 0 | 5 | 20 |
| DBLP | 28K | 68K | 300 | 0.61 | 0.76 | 4 | 1 | 1.18 | 4 | 1 | 1 | 1 |

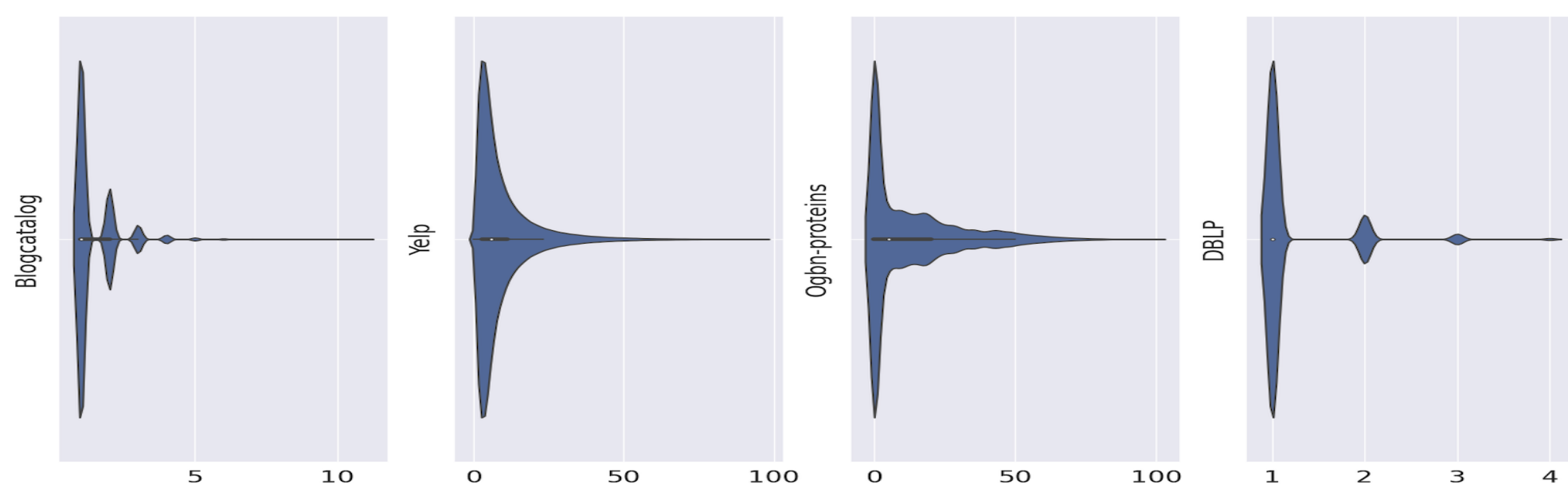

Figure 1: Label distributions. In BlogCat, the majority of the nodes have one label. In OGB-Proteins, around 41% of total nodes have no labels, and only three nodes have the maximum number of 100 labels.

**Skewed label distributions.** Figure 1 illustrates the label distributions in the four datasets. Quantitatively 72.34% nodes in BlogCat only have one label. However, the most labeled data points are assigned with 11 labels. Yelp has a total of 100 labels, the most labeled data points have 97 labels, whereas over 50% of the nodes have equal or less than 5 labels. Nevertheless, Yelp exhibits a high multi-label character with 75% of the nodes with more than 3 labels. OGB-Proteins is an extreme case in which 40.27% of the nodes do not have any label. DBLP is the dataset with the highest portion of nodes with single labels, with the exact percentage of 85.4%.

**Issue in evaluation using AUROC scores under high label sparsity.** Another so far unreported issue in multi-label datasets is the unlabeled data. In OGB-Proteins, 40.27% of nodes do not have labels. Moreover, 89.4% of the test nodes are unlabelled. More worrying is the use of the AUROC score metric in the OGB leaderboard to benchmark methods for multi-label classification. In particular, a model that assigns "No Label" to each node (i.e. predict negative class corresponding to each of the independent $L$ binary classification tasks) will already show a high AUROC score. We in fact observed that increasing the number of training epochs (which encourage the model to decrease training loss by predicting the negative class) increased the AUROC score whereas other metrics like AP or F1 score dropped or stayed unchanged.

**Cross-class neighborhood similarity.** In Figure 2 we visualize the cross-class neighborhood similarity matrix for DBLP and BlogCat. The cells on the diagonal reflect the intra-class neighborhood similarity, whereas the other cells indicate the inter-class neighborhood similarity computed using equation 1. The contrast in 2a means that nodes from the same class tend to have similar label distribution in their neighborhood, while nodes from different classes have rather different label distributions in their neighborhoods. We will later see in the experimental section that GNNs indeed benefit from this characteristic to identify correctly the nodes in the same classes in DBLP. On the contrary, the intra- and inter-class similarity are more similar in BlogCat, making it intricate for GNNs to classify the nodes to their corresponding classes.

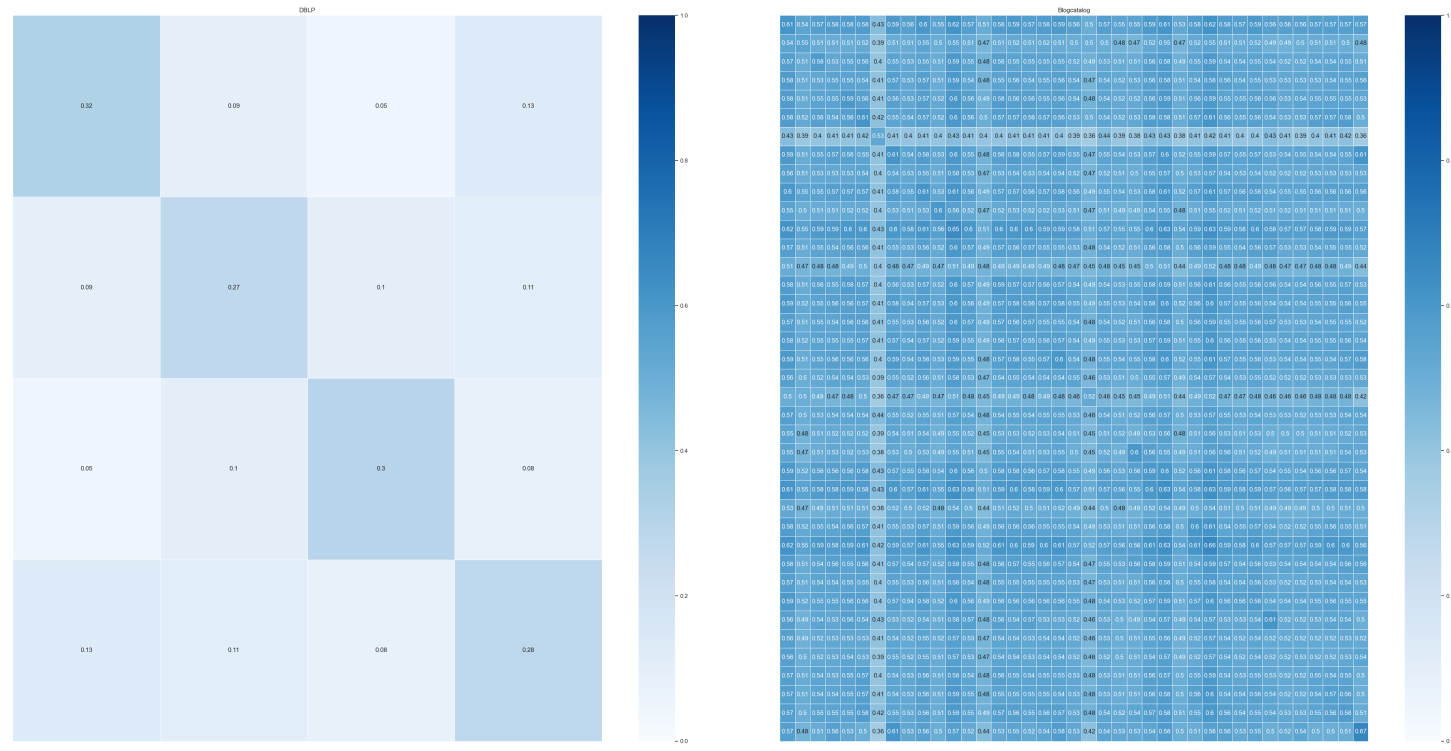

(a) Cross class Neighborhood Similarity in DBLP

(b) Cross class Neighborhood Similarity in BlogCat

Figure 2: Cross class Neighborhood Similarity in real-world datasets

### 3.2 New biological interaction datasets

Motivated by the natural applicability of the multi-label classification task in various biological datasets and to improve the representativeness of available datasets, we collect three real-world biological datasets corresponding to different multi-label classification problems: the PCG dataset for the protein phenotype prediction, the HumLoc, and EukLoc datasets for the human and eukaryote protein subcellular location prediction tasks, respectively. On each dataset, we build a graph in which each protein is modeled as a node. The node label is the corresponding protein's label. An edge represents a known interaction between two proteins retrieved from a public database. The detailed pre-processing steps and the original data sources are discussed in Appendix A.1.1, A.1.2, and A.1.3. Table 2 presents an overview of the three datasets' characteristics.

Table 2: Statistics for new datasets. The column notations are the same as in Table 1.

| Dataset | $\lvert\mathcal{V}\rvert$ | $\lvert\mathcal{E}\rvert$ | $\lvert\mathcal{F}\rvert$ | $clus$ | $r_{homo}$ | $C$ | $\ell_{med}$ | $\ell_{mean}$ | $\ell_{max}$ | 25% | 50% | 75% |
|---|---|---|---|---|---|---|---|---|---|---|---|---|
| PCG | 3K | 37k | 32 | 0.34 | 0.17 | 15 | 1 | 1.93 | 12 | 1 | 1 | 2 |
| HumLoc | 3.10k | 18K | 32 | 0.13 | 0.42 | 14 | 1 | 1.19 | 4 | 1 | 1 | 1 |
| EukLoc | 7.70K | 13K | 32 | 0.14 | 0.46 | 22 | 1 | 1.15 | 4 | 1 | 1 | 1 |

While all the existing datasets had very low homophily, HumLoc, and EukLoc show higher homophily. Moreover, these datasets improve the representativeness in terms of varying graph structure (reflected in computed clustering coefficient) and in node, edge, and feature sizes. On the downside, these datasets also show a similar low multi-label character, with the majority of nodes in these datasets still having a single label. Among the three datasets, PCG shows a bit more balanced label distribution (see Figure 3) as compared to the other two. Figure 4 provides the cross-class neighborhood similarity scores. All three datasets show different patterns according to CCNS measure which is desirable to analyse the differences in method's performance. While in PCG we see an overall high scores for CCNS, the difference in inter- and intra- class similarities is not prominent. HumLoc shows a slightly more contrasting intra- and inter-class neighborhood similarity. EukLoc, on the other hand, show very small neighborhood label similarities for nodes of same or different classes.

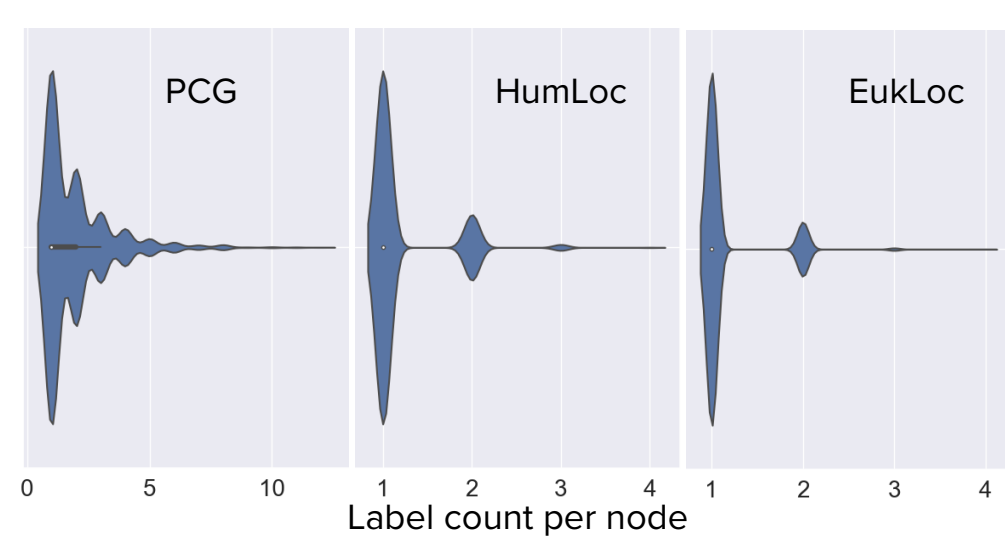

Figure 3: Label distributions in biological datasets. The majority of the nodes in all datasets have one label.

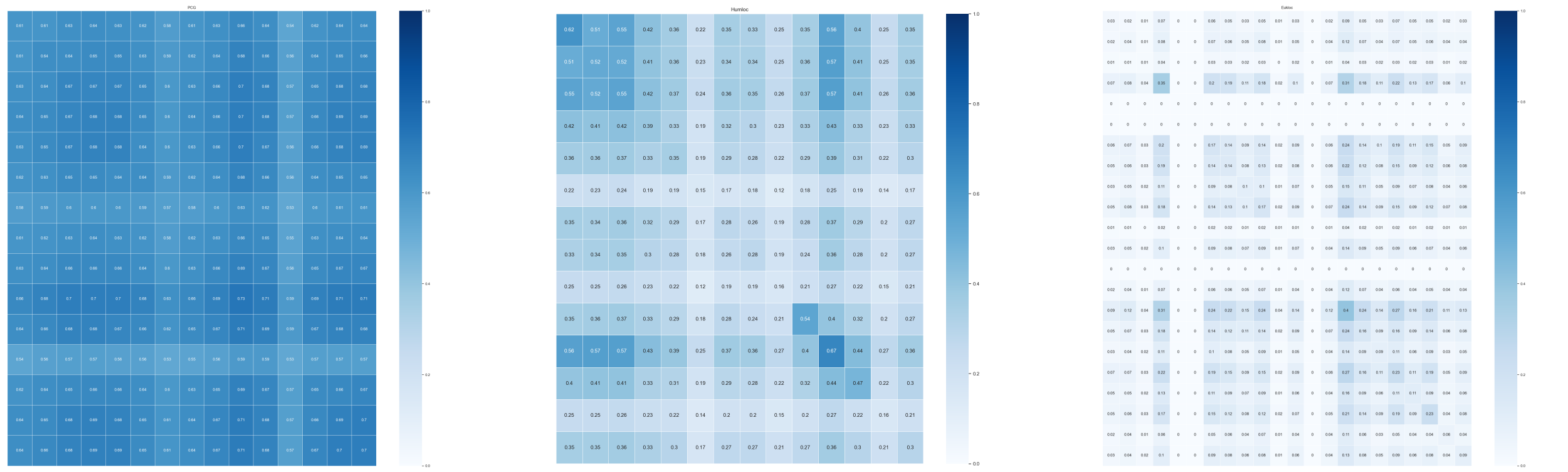

(a) Cross class Neighborhood Similarity in PCG

(b) Cross class Neighborhood Similarity in HumLoc

(c) Cross class Neighborhood Similarity in EukLoc

Figure 4: Cross class Neighborhood Similarity in real-world datasets and proposed biological datasets

## 4 Multi-label Graph Generator Framework

In the previous sections, we analyzed various real-world dataset properties which could influence a method's performance. We now develop a multi-label graph generator that will allow us to build datasets with tunable properties for a holistic evaluation. With our proposed framework we can build datasets with *high multi-label character, varying feature quality, varying label homophily, and CCNS similarity.* We now describe the two main steps of our multi-label graph generator.

**Multi-label generator.** In the first step, we generate a multi-label dataset using Mldatagen (Tomás et al., 2014). We start by fixing the total number of labels and features. We then construct a hypersphere, $H \in \mathbb{R}^{|\mathcal{F}|}$ centered at the origin and has a unit radius. Corresponding to each label in set $\mathcal{L}$ we then generate a smaller hypersphere with a random radius but with the condition that it is contained in $H$. We now start populating the smaller hyperspheres with randomly generated datapoints with $|\mathcal{F}|$ dimensions. Note that each datapoint may lie in a number of overlapping hyperspheres. The labels of the datapoint then correspond to the hyperspheres it lies in.

**Graph generator.** Having constructed the multi-label dataset, we now construct edges between the data points by using a social distance attachment model (Boguná et al., 2004). In particular, for two given datapoints (nodes) $i$ and $j$, their corresponding feature vectors are their coordinates given $\mathbf{x}_i$ and $\mathbf{x}_j$ respectively. The corresponding label vectors are denoted by $\mathbf{y}_i$ and $\mathbf{y}_j$. We denote the hamming distance between the label vectors of nodes $i$ and $j$ by $d(\mathbf{y}_i, \mathbf{y}_j)$. We then construct an edge between datapoints (nodes) $i$ and $j$, $(i, j)$ with probability given by

$$p_{ij} = \frac{1}{1 + [b^{-1} d(\mathbf{y}_i, \mathbf{y}_j)]^{\alpha}} \tag{2}$$

where $\alpha$ is a homophily parameter, $b$ is the characteristic distance at which $p_{ij} = \frac{1}{2}$. Note that the edge density is dictated by both the parameters $\alpha$ and $b$. A larger $b$ would result in denser graphs. A larger homophily parameter $\alpha$ would assign a higher connection probability to the node pairs with shorter distances or nodes with similar labels. In particular, it is a random geometric graph model, which in the limit of large system size (number of nodes) and high homophily (large $\alpha$) leads to sparsity, non-trivial clustering coefficient and positive degree assortativity (Talaga & Nowak, 2020), properties exhibited by real-world networks. By using different combinations of values of $\alpha$ and $b$, we can control the connection probability and further the label homophily of the generated synthetic graphs. We perform an extensive empirical analysis to study the relationship between $\alpha$, $b$, and the label homophily of the generated datasets. We provide detailed instructions to use our synthetic data generator and our empirical analysis in Appendix A.2.

**Synthetic datasets with fixed homophily and varying feature quality**. For our experimental analysis, we generate synthetic datasets with $3K$ nodes, 10 features, and a total of 20 labels. Towards analyzing the

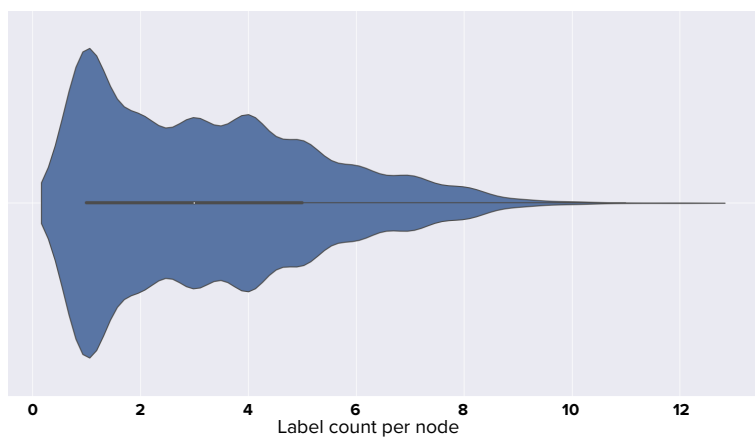

(a) Label distribution in the synthetic dataset is more balanced. A relatively high multi-label character is exhibited with 50% nodes having more than 3 labels.



(b) Cross-class Neighborhood Similarity in graph label homophily=0.2

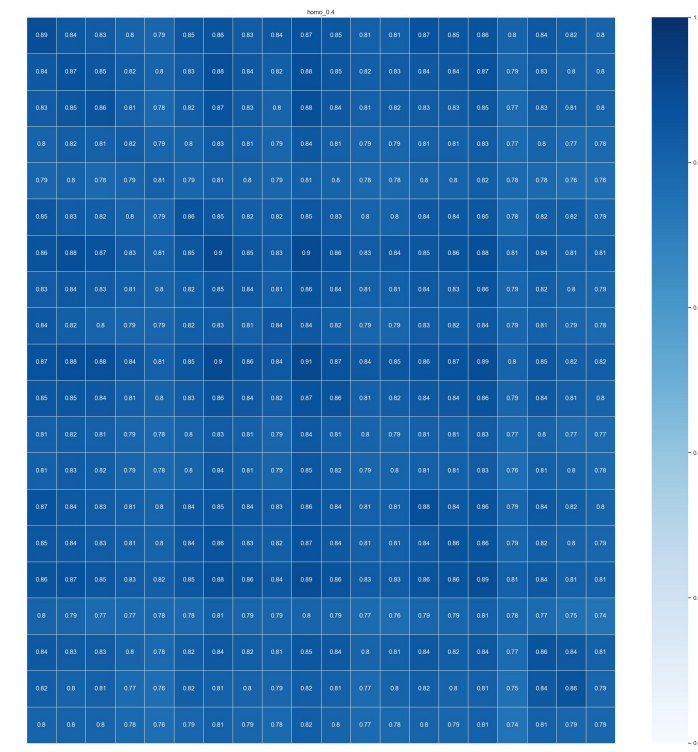

(c) Cross-class Neighborhood Similarity in graph label homophily=0.4

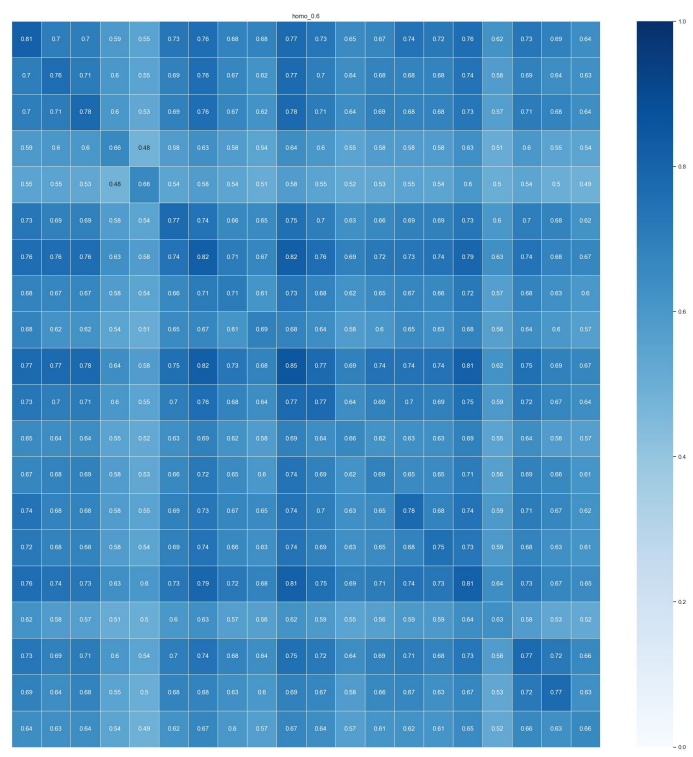

(d) Cross-class Neighborhood Similarity in graph label homophily=0.6

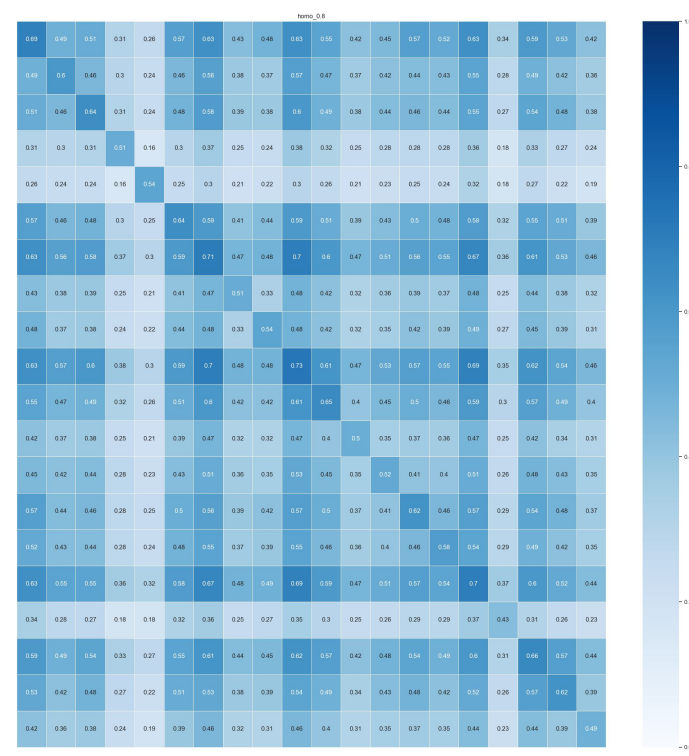

(e) Cross-class Neighborhood Similarity in graph label homophily=0.8

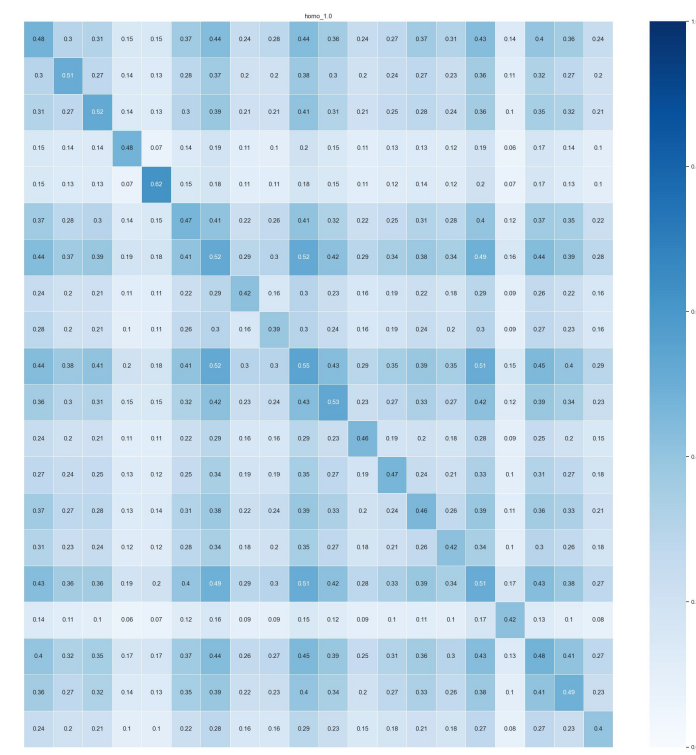

(f) Cross-class Neighborhood Similarity in graph label homophily=1.0

Figure 5: Cross-class Neighborhood Similarity in hypersphere datasets with varying label homophily

variation in the method's performance with variation in *homophily* and *feature quality* first we constructed a dataset with fixed homophily of 0.377 (using $\alpha = 8.8$, $b = 0.12$) and edge set of size $1M$. We refer to this dataset as Synthetic1. We create five variants of Synthetic1 with varying feature quality. In particular, we add 10 random features for every node, which we refer to as *irrelevant* features. We then generate its variants by removing original features such that the ratio of the number of original to that of irrelevant features varies as in $\{1, 0.8, 0.5, 0.2, 0\}$. From the label distribution plot in 5a, we observe the dataset is more multi-label than the real-world datasets because a higher number of nodes now have multiple labels.

**Synthetic datasets with varying homophily and CCNS**. We also use 5 different pairs of $\alpha$ and $b$ and the same multi-label data from the first step to construct 5 synthetic graphs with label homophily (rounded up) in $\{0.2, 0.4, 0.6, 0.8, 1.0\}$ to conduct the experiment where we test the influence of the label homophily on the performances of the node classification methods. The detailed statistics of the synthetic datasets are provided in Appendix A.2.1 in Table 5. Figure 5 visualizes the cross-class neighborhood similarity in the five hypersphere datasets with varying homophily levels (homophily varies in [0.2, 0.4, 0.6, 0.8, 1.0]). In the synthetic graphs with low label homophily, the intra- and inter-class neighborhood similarities show no significant differences, i.e., the nodes from different classes have similar label distributions in their neighborhoods. We overall observe a high absolute value of neighborhood similarity. The reason here is that similar to BlogCat (which also shows overall high CCNS scores) these synthetic low homophily graphs are highly connected and have a high average degree. As the label homophily gets higher, the contrast between the intra- and inter-class similarity becomes more significant.

## 5 Experiments

From our dataset analysis in the previous sections we observe that, unlike commonly used multi-class datasets, multi-label datasets usually have low label homophily and a more varied cross-class neighborhood similarity. Moreover, similar to the case of multi-class datasets, node features might not always be available (as in BlogCat) or might be noisy. In this section, we perform a large-scale empirical study comparing 8 methods over 7 real-world multi-label datasets and 2 sets of synthetic datasets with varying homophily and feature quality. Our experiments are designed to reveal and understand (i) properties of datasets that favor certain methods over others and (ii) the effect of varying feature quality and label homophily on method performance. The training and hyperparameter settings for each model are summarized in the Appendix A.3 in tables 7 and 8. Our code is available at `https://github.com/Tianqi-py/MLGNC`.

**Datasets.** We employ 7 real-world datasets including BlogCat, Yelp, OGB-Proteins, DBLP, PCG, HumLoc, EukLoc, and 2 sets of synthetic datasets with varying homophily and feature quality. These datasets are already described in Section 3. For all datasets except OGB-Proteins, HumLoc, and EukLoc we generate 3 random training, validation, and test splits with 60%, 20%, and 20% of the data. For OGB-Proteins, HumLoc, and EukLoc we follow the predefined data splits from (Hu et al., 2020), (Shen & Chou, 2007) and (Chou & Shen, 2007) respectively. As BlogCat has no given node features, we use an identity matrix as the input feature matrix.

**Compared Methods.** For a holistic evaluation we include four classes of compared methods (i) simple methods, which include Multilayer Perceptron (Mlp), which only uses node features and ignores the graph structure, and DeepWalk, which only uses graph structure and ignores the node features (ii) Convolutional neural networks based which employ convolutional operations to extract representations from node's local neighborhoods and merge them with label embeddings for the final classification. We choose LANC as a baseline from this category, as previous works (Zhou et al., 2021; Song et al., 2021) have shown its superior performance. (iii) graph neural networks including (a) Gcn, Gat, and GraphSage, which are known to perform well for graphs with high label homophily, and (b)H2Gcn, which is designed to perform well both on homophilic and heterophilic graphs and (iiii) GCN-LPA which combines label propagation and GCN for node classification. All these methods are also discussed in Section 2.

**Evaluation Metrics.** We report the average micro- and macro-F1 score, macro-averaged AUC-ROC score, macro-averaged average precision score, and standard deviation over the three random splits. Due to space constraints, we report average precision (AP) in the main paper, and all detailed results are available in Tables 9, 10, 11 in the Appendix A.4. Our choice of using AP over AUROC as the metric is also motivated in Appendix A.5.

## 6 Results and Discussion

Table 3: Mean performance scores (Average Precision) on real-world datasets. The best score is marked in bold. The second best score is marked with underline. "OOM" denotes the "Out Of Memory" error.

| Method | BlogCat | Yelp | OGB-Proteins | DBLP | PCG | HumLoc | EukLoc |
|---|---|---|---|---|---|---|---|
| Mlp | 0.043 | 0.096 | 0.026 | 0.350 | 0.148 | 0.170 | 0.120 |
| DeepWalk | **0.190** | 0.096 | 0.044 | 0.585 | **0.229** | 0.186 | 0.076 |
| LANC | 0.050 | OOM | 0.045 | 0.836 | 0.185 | 0.132 | 0.062 |
| Gcn | 0.037 | 0.131 | **0.054** | **0.893** | 0.210 | **0.252** | **0.152** |
| Gat | 0.041 | 0.150 | 0.021 | 0.829 | 0.168 | 0.238 | 0.136 |
| GraphSage | 0.045 | **0.251** | 0.027 | 0.868 | 0.185 | 0.234 | 0.124 |
| H2Gcn | 0.039 | 0.226 | 0.036 | 0.858 | 0.192 | 0.172 | 0.134 |
| GCN-LPA | 0.043 | 0.116 | 0.023 | 0.801 | 0.167 | 0.150 | 0.075 |

### 6.1 Results on real-world datasets.

Table 3 we provide the results for 7 real-world datasets. In general, on datasets with low label homophily, such as BlogCat and PCG, node representation or embedding learning methods such as DeepWalk, outperform more sophisticated GNN based methods and simple Mlp baseline. Classical GNNs show better performance on datasets characterized by high label homophily. H2Gcn which is designed for multi-class datasets with heterophily do not show a performance improvement over classical GNNs on multi-label graph datasets with low homophily. Likewise, the method that combine label propagation with GNNs, achieve only comparable results to classical GNNs.

**BlogCat.** For BlogCat all GNN approaches as well as LANC and GCN-LPA obtain scores close to that of Mlp which do not use any graph structure. Notably, Mlp uses identity matrix as input features which in principle provides no useful information. The corresponding scores can be seen as the result of a random assignment. As a method unifying the label and feature propagation, GCN-LPA does not show improvement on BlogCat compared to other baselines. This is because GCN-LPA uses the label propagation to adjust the edge weight and still only generates embedding by aggregating features over the weighted graph, while DeepWalk take advantage of the informative topological structure in the graph and achieves the best performance.

**Yelp, OGB-Proteins and DBLP.** Yelp has low label homophily and low clustering coefficient but a more balanced label distribution as compared to other real-world datasets, so approaches designed specifically for low homophilic graphs and are capable of preserving information from both low- and high- order neighborhoods are expected to perform better for this dataset. Among the GNN-based baselines, H2Gcn which has shown improvements in low homophilic multi-class datasets outperforms Gcn,Gat, and GCN-LPA but is still outperformed by GraphSage. The use of CNNs in LANC to aggregate neighborhood features leads to excessive memory utilization for a graph with a very high maximum degree. This led to the out-of-memory error for Yelp.

All methods perform poorly on OGB-Proteins with Gcn and LANC slighty outperforming others. It also has low homophily which is similar to BlogCat, which also does not provide GNNs with any additional advantage. In particular, DeepWalk and H2Gcn outperform Gat and GraphSage as well as more sophisticated GCN-LPA. It is worth noting that the previously reported results on the OGB-Proteins leaderboard Hu et al. (2020) are significantly exaggerated due to the utilization of the AUROC metric in conjunction with excessive model training. When using the metic Average Precision, the scores we get is much lower that what were reported on the leaderboard.

As a co-authorship dataset, DBLP has the highest label homophily and the largest portion of nodes with a single label among all the real-world datasets. As shown in Figure 2a in the section 3.1, the inter-class similarity is much weaker than the intra-class similarity. Besides, the high clustering coefficient indicates that the local neighborhood is highly connected. All of these factors further justify the best performance shown by Gcn. Besides, the relatively poor performances of Mlp and DeepWalk indicate that the features or the structure alone are not sufficient for estimating node labels.

**PCG, HumLoc and EukLoc.** If the features are highly predictive of the labels, the simple baselines Mlp using only feature information would be a competitive baseline. In the experiments with the datasets, where the features are highly correlated (evident from relatively better performance of Mlp) with the assigned labels, i.e., the biological interaction datasets HumLoc and EukLoc (shown in Table 3), GNNs and GCN-LPA tend to have better performance than DeepWalk and LANC which only utilizes graph structure and features from the direct neighborhood. PCG exhibits low label homophily and high clustering coefficient. Consistent with observations in other low homophilic datasets DeepWalk outperforms other methods in PCG too. H2Gcn, on the other hand, is outperformed by simpler GNN baseline like GCN.

### 6.2 Results on synthetic datasets

Table 4 provides results for synthetic datasets with varying feature quality and label homophily. We provide a detailed analysis and argue about performance differences in the following sections.

Table 4: Average Precision (mean) on the synthetic datasets with varying levels of feature quality and homophily parameter. $r_{ori_feat}$ and $r_{homo}$ refer to the fraction of original features and the homophily parameter value, respectively.

| Method | $r_{ori_feat}$ | | | | | $r_{homo}$ | | | | |
|---|---|---|---|---|---|---|---|---|---|---|
| | 0.0 | 0.2 | 0.5 | 0.8 | 1.0 | 0.2 | 0.4 | 0.6 | 0.8 | 1.0 |
| Mlp | 0.172 | 0.187 | 0.220 | 0.277 | 0.343 | **0.343** | 0.343 | 0.343 | 0.343 | 0.343 |
| DeepWalk | **0.487** | **0.487** | **0.487** | **0.487** | **0.487** | 0.181 | **0.522** | **0.813** | **0.869** | 0.552 |
| LANC | 0.337 | 0.342 | 0.365 | 0.353 | 0.391 | 0.190 | 0.380 | 0.434 | 0.481 | 0.629 |
| Gcn | 0.313 | 0.316 | 0.311 | 0.301 | 0.337 | 0.261 | 0.343 | 0.388 | 0.450 | 0.493 |
| Gat | 0.311 | 0.339 | 0.329 | 0.338 | 0.360 | 0.172 | 0.359 | 0.390 | 0.428 | 0.439 |
| GraphSage | 0.300 | 0.328 | 0.377 | 0.393 | 0.430 | 0.289 | 0.426 | 0.458 | 0.533 | 0.553 |
| H2Gcn | 0.376 | 0.401 | 0.427 | 0.442 | 0.467 | 0.297 | 0.484 | 0.512 | 0.572 | **0.652** |
| GCN-LPA | 0.337 | 0.333 | 0.368 | 0.363 | 0.391 | 0.170 | 0.408 | 0.495 | 0.604 | 0.583 |

#### 6.2.1 Effect of varying feature quality.

As simple baseline Mlp and other GNN-based models use features as input, we assume they will be sensitive to the varying feature quality. As DeepWalk uses the graph structure alone, its performance would not be affected by the varying feature quality. To further validate our hypothesis and test the robustness of the methods to feature quality, we compare the method performances on variants of the generated Synthetic1 dataset. Specifically, we vary the ratio of the original to the irrelevant features as in $\{0, 0.2, 0.5, 0.8, 1.0\}$.

As shown in Table 4, under all levels of feature quality, the performances of DeepWalk do not change, as it generates representations for the nodes solely from the graph structure. The Mlp is unsurprisingly the most sensitive method to the varying feature quality because it completely ignores the graph structure. LANC is also sensitive to the change of the feature quality as it extracts feature vectors from the local neighborhood by performing convolutional operations on the stacked feature matrix of the direct neighbors.

The Synthetic1 dataset used in this experiment has a label homophily of 0.3768, which is a relatively high label homophily for the multi-label datasets. Surprisingly, GCN-LPA which employs the label information performs only a little better than GraphSage on the feature-to-noise ratio of 0 and 0.2. H2Gcn, on the other hand, outperforms all GNN-based baselines and GCN-LPA for all levels of feature quality.

#### 6.2.2 Effect of varying label homophily.

In this subsection, we test the robustness of the methods to varying homophily and further argue how low label homophily has different semantics on multi-label graphs as it does on multi-class datasets. Specifically, we vary the label homophily as in $\{0.2, 0.4, 0.6, 0.8, 1.0\}$.

As shown in Table 4, the performances of Mlp do not change under different levels of label homophily, due to the fact that Mlp only use the input features.

H2Gcn is a method that has been shown to perform well on heterophilic multi-class datasets. In the multi-label scenarios, it exhibits better performance than other GNN methods but is outperformed by simple MLP baseline in case of label homophily of 0.2.

On the other hand, with most of the attention drawn to developing new complicated methods for the node classification task, we observe simple baselines such as DeepWalk outperform standard GNNs in several scenarios. On the synthetic datasets with the label homophily of 0.4, 0.6 and 0.8, DeepWalk is the best-performing method. As shown in Table 4, the performance of DeepWalk drops when the label homophily is 1.0. However, we want to emphasize that this is because we fixed the same walk length for DeepWalk for all levels of label homophily and the improvement can be shown with a possible hyperparameter tuning.

As mentioned in section 4, the intra-class similarity is significantly stronger than the inter-class similarity in the synthetic graphs with higher label homophily, which helps GNN-based models to better distinguish the nodes from different classes and thus achieve better results in the node classification task.

## 7 Conclusion

We investigate the problem of multi-label node classification on graph-structured data. Filling in the gaps in current literature, we (i) perform in-depth analysis on the commonly used benchmark datasets, create and release several real-world multi-label datasets and a graph generator model to produce synthetic datasets with tunable properties, (ii) compare and analyse the performances of the methods from different categories for the node classification task by conducting large-scale experiments on 9 datasets and 8 methods, and (iii) release our benchmark publicly.

We have novel and compelling insights from our analysis of specific datasets and GNN approaches. For instance, we uncover the pitfalls of the commonly used OGB-Protein dataset for model evaluation. While multi-label graph datasets usually show low homophily, we show that approaches working on low homophilic multi-class datasets cannot trivially work on multi-label datasets which usually have low homophily.

While current graph-based machine learning methods are usually evaluated on multi-class datasets, we demonstrate that the acquired improvements cannot always be translated to the more general scenario when the nodes are characterized by multiple labels. We believe that our work will open avenues for more future work and bring much-deserved attention to multi-label classification on graph-structured data. In future work, we plan to study the interplay of different dataset characteristics (for example edge density and label homophily) on the model performance.

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

# A Appendix

**Organization.** We explain the construction details of the biological datasets in A.1. Furthermore, in Section A.2, we study the parameters of the graph generator and demonstrate how we generate the synthetic datasets with varying label homophily. We also summarize the characteristics of the generated synthetic graphs that were used in Section 5 for the varying homophily and feature quality experiments. In Section A.3, we summarize the hyperparameters of the models we used in this work. In Section A.4, we provide the full original experiment results on all datasets reported in Micro- and Macro- F1, AUROC, and Average Precision score with the standard deviation of the 3 random splits if the dataset is not pre-splitted. Note that for higher precision, the scores are provided in percentages. Last but not least, we provide the motivation for using Average Precision in the main paper in Section A.5.

### A.1 Biological Dataset Construction

#### A.1.1 The Protein phenotype prediction dataset.

A phenotype is any observable characteristic or trait of a disease. Identifying the phenotypes associated with a particular protein could help in clinical diagnostics or finding possible drug targets.

To construct the phenotype prediction dataset, we first retrieve the experimentally validated protein-phenotype associations from the DisGeNET (Piñero et al., 2020) database. We then (i) retain only those protein associations that are marked as "phenotype", (ii) match each disease to its first-level category in the MESH ontology (Bhattacharya et al., 2011), and (iii) remove any (phenotype) label with less than 100 associated proteins. To construct the edges, we acquire the protein functional interaction network from (Wu et al., 2010) (version 2020). We then (i) model each protein as a node in the graph, (ii) retain only the protein-protein interactions between the proteins that have the phenotype labels available, and (iii) remove any isolated nodes from the constructed graph. In the end, our dataset consists of $3,233$ proteins and $37,351$ edges. The node features are the 32-dimensional sequence-based embeddings retrieved from (Consortium, 2015) and (Yang et al., 2020).

#### A.1.2 The human protein subcellular location prediction dataset (HumLoc).

Proteins might exist at or move between different subcellular locations. Predicting protein subcellular locations can aid the identification of drug targets[1]. We retrieve the human protein subcellular location data from (Shen & Chou, 2007) which contains $3,106$ proteins. Each protein can have one to several labels in 14 possible locations. We then generate the graph multi-label node classification data as follows:

- We model each protein as a node in the graph. We retrieve the corresponding protein sequences from Uniprot (Consortium, 2015). We obtain the corresponding 32-dimensional node feature representation by feeding them to a pre-trained model (Yang et al., 2020) on protein sequences.
- Each node's label is the one-hot encoding (i.e., 14 dimensions) generated from its sub-cellular information. Each value in the label vector represents one sub-cellular location. A value of 1 indicates the corresponding protein exists at the respective location and 0 means otherwise.
- The edge information is generated from the protein-protein interactions retrieved from the IntAct (Kerrien et al., 2012) database. There exists a connection between two nodes in the graph if there exists an interaction between the corresponding proteins in IntAct. For each pair of proteins, more than one interaction of different types might exist. Therefore, we assign each edge a label. The edge label is modeled as a 21-dimensional vector where each value in the vector represents the confidence score for a particular connection type.

In the end, the HumLoc dataset consists of 3,106 nodes and 18,496 edges. Each node can have one to several labels in the 14 possible locations. Among the 3,106 different proteins, 2,580 of them belong only to 1 location; 480 of them belong to 2 locations; 43 of them belong to 3 locations and 3 of them belong to 4 locations. Both the accession numbers and sequences are given. None of the proteins has more than 25% sequence identity to any other in the same subset (subcellular location). For a more detailed description of the original dataset, we refer the readers to (Shen & Chou, 2007).

#### A.1.3 The eukaryote protein subcellular location prediction dataset (EukLoc).

We retrieve the eukaryote protein subcellular location multi-label data from (Chou & Shen, 2007). We then employ the same data sources and pre-processing strategy as described for the HumLoc dataset to generate the multi-label node classification dataset for eukaryote protein subcellular location prediction. In the end, the final pre-processed data contains 7,766 proteins (nodes) and 13,818 connections(edges). Each protein(node) can receive one to several labels in 22 possible locations.

[1] https://en.wikipedia.org/wiki/Protein_subcellular_localization_prediction

### A.2 Parameter Study of the Graph Generator Model

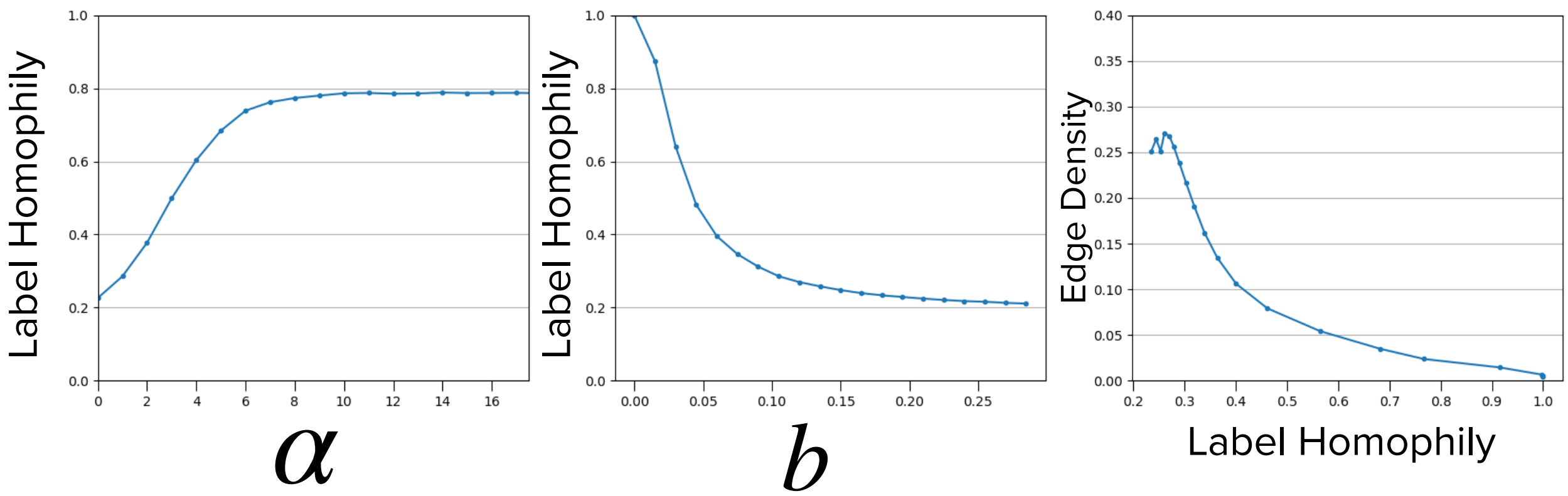

Figure 6: Visualization of the parameter study of the Graph Generator Model. The first two subplots demonstrate the relationships between the value of $\alpha$ and $b$ and the label homophily of the generated synthetic datasets. The last subplot shows the edge density and the label homophily of the generated synthetic graphs. This shows with the same multi-label data, we can generate synthetic graphs with varying label homophily.

As mentioned in Section 4, the choice of $\alpha$ and $b$ will directly determine the connection probability $p_{i,j}$ of each pair of nodes $i$ and $j$ and the homophily ratio of the generated synthetic graph. Here, we demonstrate how we choose the value of $\alpha$ and $b$ to generate synthetic graphs with varying homophily ratios. Note that the valid range of $\alpha$ and $b$ may differ when a different distance metric is used.

Firstly, we randomly choose 500 nodes from Synthetic1 dataset and generate a series of small synthetic graphs with varying $\alpha$ and $b$ and observe the relationship between them. Since we have two hyperparameters' ranges to determine, we first fix the value of one and explore the range of the other and then vice versa. To recall, $b$ indicates the characteristic distance at which $p_{ij} = \frac{1}{2}$, and our hamming distance should be in the range of $[0, 1]$, we first fix $b$ to 0.05. We chose a small value of $b$ because the larger value of $b$ would dominate the influence of $\alpha$ and the relationship between the homophily ratio and the change of the value of $\alpha$ becomes unclear. The relationship between the homophily ratio of the generated synthetic graphs and the value of $\alpha$ is shown in Figure 6(a). As shown in the subplot, the label homophily increases monotonically as the value of $\alpha$ increases in the range of $[0, 10]$. As $\alpha$ is interpreted as the homophily parameter in (Boguná et al., 2004), only positive values make sense.

Similarly, we then fix $\alpha$ to its middle value in the valid range, i.e. 5, and explore the valid range of $b$ and visualize the relationship of the graph level homophily ratio and the value of $b$ in Figure 6(b). As illustrated in the subfigure, the label homophily decreases as the b increases in the range of $(0, 0.25]$. As $b$ increases, the node pairs with bigger distance would also have 50% of the probability to be connected, the number of edges will increase, and the label homophily ratio will decrease. As $b$ decreases, the node pair with a smaller distance would only have 50% of the probability of being connected. The model becomes cautious about connecting a node pair. The number of edges decreases and only the node pairs, which are alike will be connected, thus, the label homophily ratio increases.

Then, we use combinations of $\alpha$ and $b$ to generate synthetic graphs with specific homophily ratios. We sample 20 $\alpha$s and $b$s uniformly from their valid ranges with the increments 0.5 and 0.0125. Since the graph label homophily has an inverse linear relationship with $\alpha$ and $b$, we arrange the sampled $b$ in reverse order and then form 20 value pairs (alpha, b). We generate 20 synthetic graphs from the multi-label dataset corresponding to Synthetic1 with these value pairs ($\alpha$, $b$) and plot the homophily ratio and the edge density of the generated graphs in Figure 6(c). As shown in the subplot, using the same multi-label data, we are able to generate synthetic graphs with varying homophiles. And the edge density decreases when label homophily increases. As in higher label homophily graphs, the generator will only connect the nodes that are highly similar to each other. In contrast, when the label homophily is low, the graph generator will connect every possible node pair in the graph resulting in denser graphs.

### A.2.1 Statistics Of The Synthetic Datasets

Here we summarize the characteristics of the generated synthetic graphs with varying label homophily and feature quality. The first row denotes the name of the synthetic graphs, where in the varying homophily experiments, the variants of datasets are named with their label homophily. The Synthetic1 dataset is used in the varying feature quality experiment, where we remove the relevant features to create dataset variants with varying feature quality levels. Note that for all the datasets the label distribution stays the same as they are just different graphs generated from the same multi-label dataset. The statistics on label distribution for these datasets are given in Table 5 and Table 6.

Table 5: The number of edges and clustering coefficient of the synthetic datasets with varying label homophily and Synthetic1. The row of '|E|' denotes the number of edges and the 'clustering coefficient' denotes the clustering coefficient of these datasets

| Dataset | 0.2 | 0.4 | 0.6 | 0.8 | 1.0 | Synthetic1 |
|---|---|---|---|---|---|---|
| \|E\| | 2.37M | 598k | 298k | 79.5k | 47.6k | 1.00M |
| Clustering Coefficient | 0.53 | 0.37 | 0.39 | 0.49 | 0.93 | 0.57 |

Table 6: The label distribution of the synthetic dataset used in this work. The column notations are same as in Table 1.

| | $\lvert L \rvert$ | $\lvert L_{med} \rvert$ | $\lvert L_{mean} \rvert$ | $\lvert L_{max} \rvert$ | 25% | 50% | 75% |
|---|---|---|---|---|---|---|---|
| label distribution | 20 | 3 | 3.23 | 12 | 1 | 3 | 5 |

## A.3 Hyperparameter Setting

In this section, we summarize all the hyperparameters we used for the experiment section. The detailed setting is listed in Table 7 and 8.

More specifically, for Mlp and all GNN-based methods, we summarize the number of layers, the dimension of the hidden layer, the learning rate, the patience of the Earlystopping, the weight decay, and the number of neighbors we sample for the models that require sampling.

We use the same number of layers and the same hidden size for Mlp and the other GNN-based methods. The learning rate for the synthetic datasets in the varying feature quality and homophily experiments is 0.001 instead of 0.01 as in the other models because the performance of the H2Gcn is further improved. We also use Earlystopping with the patience of 100 epochs to train the models properly. For GraphSage, we sample 25 and 20 one and two hops away neighbors for aggregation. As other GNN-based baselines do not use the sampling method, the corresponding cells are filled with "No".

For the only random-walk-based method, we deploy the default setting for all the datasets in this work as DeepWalk already shows competitive performance. We perform 10 random walks with the walk length of 10 for each node to generate the sequence and use the window size of 5 for the training pairs, the generated embedding size is 64.

## A.4 The experiment results reported in four metrics

We summarize the experimental results on the real-world datasets and the synthetic datasets in Table 9, Table 10, and Table 11, respectively. For better precision, we report the scores in percentages. Specifically, for the scores reported on OGB-Proteins, the difference between our results and those reported in the

Table 7: The hyperparameter setting for Mlp and GNN baselines in this work for all datasets

| | MLP | GCN, GAT, GraphSAGE | H2GCN | GCN-LPA |
|---|---|---|---|---|
| Layers | 2 | 2 | 2 | 2 GCN
5 LPA |
| Hidden size | 256 | 256 | 256 | 256 |
| Learning rate | 0.01 | 0.01 | real-world: 0.01
synthetic: 0.001 | 0.01 |
| Earlystopping patience | 100 | 100 | 100 | 100 |
| Weight decay | 5e-4 | 5e-4 | 5e-4 | 1e-4 |
| Sample for aggregation | No | GraphSAGE:[25, 10] | No | No |

Table 8: The hyperparameter setting for DeepWalk in this work for all datasets

| | Number of walks | Walk length | Embedding size | Window size |
|---|---|---|---|---|
| DeepWalk | 10 | 10 | 64 | 5 |

benchmark is because 1) we use 2 layer Mlp without Node2Vec features. 2) We use Earlystopping with the patience of 100 epochs to prevent the models from being overtrained. 3) We use sampled local neighborhoods to have a consistent setting for all the datasets using GraphSage. The specific parameters we used are summarized in A.3. Another pre-processing we did was to remove the isolated nodes in PCG. The details are described in Appendix A.1.1.

Table 9: Multi-label Node Classification results on real-world datasets. The results of BlogCat, Yelp, PCG, and DBLP are the mean of three random splits with 60% for train-, 20% validation and 20% test dataset, while the results of OGB-Proteins, HumLoc, EukLoc are reported with the built-in split.

| **Dataset** | **Method** | Micro-F1 | Macro-F1 | AUC-ROC | AP |
|---|---|---|---|---|---|
| Blog-Cat | Mlp | $17.11 \pm 0.64$ | $2.49 \pm 0.18$ | $50.30 \pm 1.04$ | $4.25 \pm 0.63$ |
| | DeepWalk | $\mathbf{35.59 \pm 0.21}$ | $\mathbf{19.74 \pm 0.54}$ | $\mathbf{73.20 \pm 0.58}$ | $\mathbf{18.55 \pm 0.17}$ |
| | LANC | $13.95 \pm 2.02$ | $4.55 \pm 0.82$ | $52.34 \pm 0.91$ | $\underline{5.03 \pm 0.07}$ |
| | Gcn | $16.69 \pm 0.47$ | $\underline{2.63 \pm 0.08}$ | $47.85 \pm 0.06$ | $3.69 \pm 0.04$ |
| | Gat | $\underline{17.22 \pm 0.52}$ | $2.48 \pm 0.08$ | $50.88 \pm 1.45$ | $4.05 \pm 0.09$ |
| | GraphSage | $16.18 \pm 0.31$ | $2.38 \pm 0.27$ | $\underline{52.73 \pm 0.82}$ | $4.50 \pm 0.12$ |
| | H2Gcn | $16.86 \pm 0.34$ | $2.60 \pm 0.16$ | $49.83 \pm 1.08$ | $3.92 \pm 0.05$ |
| | GCN-LPA | $17.15 \pm 0.68$ | $2.55 \pm 0.19$ | $51.35 \pm 0.67$ | $4.33 \pm 0.31$ |
| Yelp | Mlp | $26.04 \pm 0.09$ | $18.55 \pm 0.20$ | $50.17 \pm 0.01$ | $9.58 \pm 0.01$ |
| | DeepWalk | $49.78 \pm 0.07$ | $24.98 \pm 0.04$ | $50.67 \pm 0.08$ | $9.60 \pm 0.02$ |
| | LANC | – | – | – | – |

Table 9: Multi-label Node Classification results on real-world datasets. The results of BlogCat, Yelp, PCG, and DBLP are the mean of three random splits with 60% for train-, 20% validation and 20% test dataset, while the results of OGB-Proteins, HumLoc, EukLoc are reported with the built-in split.

| **Dataset** | **Method** | Micro-F1 | Macro-F1 | AUC-ROC | AP |
|---|---|---|---|---|---|
| | Gcn | $52.21 \pm 0.07$ | $27.60 \pm 0.04$ | $53.81 \pm 0.13$ | $13.14 \pm 0.06$ |
| | Gat | $51.24 \pm 0.08$ | $26.66 \pm 0.06$ | $67.80 \pm 0.05$ | $15.00 \pm 0.07$ |
| | GraphSage | $\mathbf{56.06 \pm 0.10}$ | $\mathbf{31.26 \pm 0.10}$ | $\mathbf{81.05 \pm 0.25}$ | $\mathbf{25.09 \pm 0.31}$ |
| | H2Gcn | $54.12 \pm 0.01$ | $30.52 \pm 0.11$ | $75.25 \pm 0.46$ | $22.57 \pm 0.51$ |
| | GCN-LPA | $50.31 \pm 0.29$ | $25.68 \pm 0.43$ | $61.09 \pm 2.27$ | $11.62 \pm 0.74$ |
| OGB-Proteins | Mlp | 2.55 | 2.40 | 54.05 | 2.59 |
| | DeepWalk | **2.88** | **2.75** | 68.75 | 4.41 |
| | LANC | 2.35 | 2.21 | 68.03 | 4.48 |
| | Gcn | 2.77 | 2.63 | **71.48** | **5.36** |
| | Gat | 2.55 | 2.40 | 50.64 | 2.14 |
| | GraphSage | 2.59 | 2.43 | 55.83 | 2.68 |
| | H2Gcn | 2.55 | 2.39 | 62.75 | 3.61 |
| | GCN-LPA | 2.56 | 2.41 | 53.22 | 2.33 |
| DBLP | Mlp | $42.14 \pm 0.27$ | $32.04 \pm 0.65$ | $54.47 \pm 0.07$ | $34.97 \pm 0.14$ |
| | DeepWalk | $63.27 \pm 0.34$ | $59.11 \pm 0.36$ | $74.81 \pm 0.16$ | $58.49 \pm 0.25$ |
| | LANC | $81.93 \pm 0.29$ | $80.39 \pm 0.42$ | $91.76 \pm 0.36$ | $83.55 \pm 0.98$ |
| | Gcn | $\mathbf{87.03 \pm 0.20}$ | $\mathbf{85.80 \pm 0.38}$ | $94.15 \pm 0.16$ | $\mathbf{89.27 \pm 0.24}$ |
| | Gat | $83.06 \pm 0.17$ | $81.26 \pm 0.14$ | $92.57 \pm 0.07$ | $82.93 \pm 0.16$ |
| | GraphSage | $85.22 \pm 0.23$ | $83.89 \pm 0.21$ | $\mathbf{94.32 \pm 0.02}$ | $86.84 \pm 0.18$ |
| | H2Gcn | $83.99 \pm 0.92$ | $82.56 \pm 0.86$ | $92.14 \pm 0.57$ | $85.82 \pm 0.64$ |
| | GCN-LPA | $82.88 \pm 0.31$ | $81.31 \pm 0.34$ | $90.17 \pm 0.43$ | $80.07 \pm 1.24$ |
| PCG | Mlp | $38.04 \pm 1.20$ | $18.03 \pm 1.29$ | $51.07 \pm 0.63$ | $14.78 \pm 0.60$ |
| | DeepWalk | $\mathbf{42.26 \pm 1.37}$ | $\mathbf{31.49 \pm 0.90}$ | $\mathbf{63.58 \pm 0.87}$ | $\mathbf{22.86 \pm 1.00}$ |
| | LANC | $36.28 \pm 0.34$ | $20.50 \pm 1.15$ | $56.58 \pm 0.69$ | $18.53 \pm 1.14$ |
| | Gcn | $41.46 \pm 1.21$ | $25.59 \pm 0.92$ | $59.54 \pm 0.82$ | $21.03 \pm 0.34$ |
| | Gat | $36.91 \pm 1.75$ | $19.24 \pm 0.75$ | $56.33 \pm 4.64$ | $16.75 \pm 2.17$ |
| | GraphSage | $38.89 \pm 1.17$ | $24.44 \pm 1.74$ | $58.57 \pm 0.08$ | $18.45 \pm 0.29$ |
| | H2Gcn | $39.05 \pm 0.99$ | $24.38 \pm 2.17$ | $58.10 \pm 0.14$ | $19.19 \pm 0.49$ |
| | GCN-LPA | $39.57 \pm 1.12$ | $22.90 \pm 1.33$ | $54.74 \pm 0.95$ | $16.71 \pm 0.14$ |
| Hum-Loc | Mlp | 42.12 | 18.04 | 66.04 | 16.95 |
| | DeepWalk | 45.26 | 23.30 | 65.67 | 18.58 |
| | LANC | 39.25 | 11.51 | 59.65 | 13.24 |
| | Gcn | **51.67** | **25.57** | 67.28 | **25.15** |
| | Gat | 47.10 | 17.49 | **72.47** | 23.75 |
| | GraphSage | 48.05 | 21.22 | 70.30 | 23.42 |
| | H2Gcn | 45.39 | 18.35 | 64.31 | 17.23 |
| | GCN-LPA | 45.73 | 18.15 | 62.40 | 14.96 |
| Eu-k-Loc | Mlp | 43.58 | 11.13 | 66.83 | 12.00 |
| | DeepWalk | 34.67 | 6.74 | 56.12 | 7.58 |
| | LANC | 36.08 | 4.55 | 51.13 | 6.16 |
| | Gcn | **45.86** | **12.27** | 70.53 | **15.15** |
| | Gat | 41.58 | 6.76 | **71.65** | 13.59 |
| | GraphSage | 44.65 | 11.96 | 69.04 | 12.44 |
| | H2Gcn | 44.93 | 11.80 | 69.45 | 13.35 |
| | GCN-LPA | 36.72 | 5.93 | 56.65 | 7.45 |

Table 10: Multi-label Node Classification results on Synthetic dataset with varying feature quality. All results are the mean of three random splits. The Ratios of the relevant and the irrelevant features are $[0, 0.2, 0.5, 0.8, 1.0]$.

| **Feature Ratios** | **Method** | Micro-F1 | Macro-F1 | AUC-ROC | AP |
|---|---|---|---|---|---|
| 0 | Mlp | $67.05 \pm 0.91$ | $25.15 \pm 0.89$ | $50.99 \pm 1.11$ | $17.17 \pm 0.42$ |
| | DeepWalk | $\mathbf{86.22 \pm 0.39}$ | $\mathbf{47.80 \pm 0.31}$ | $\mathbf{84.15 \pm 0.56}$ | $\mathbf{48.70 \pm 0.51}$ |
| | LANC | $69.67 \pm 0.26$ | $26.68 \pm 1.16$ | $75.36 \pm 1.17$ | $33.68 \pm 0.88$ |
| | Gcn | $67.09 \pm 0.95$ | $25.12 \pm 1.09$ | $77.17 \pm 0.92$ | $31.27 \pm 0.66$ |
| | Gat | $66.67 \pm 0.71$ | $25.09 \pm 1.24$ | $61.96 \pm 3.97$ | $31.10 \pm 3.17$ |
| | GraphSage | $67.90 \pm 1.40$ | $25.87 \pm 1.51$ | $66.47 \pm 0.31$ | $30.01 \pm 0.71$ |
| | H2Gcn | $\underline{71.12 \pm 1.26}$ | $\underline{27.45 \pm 1.18}$ | $\underline{80.83 \pm 0.98}$ | $\underline{37.64 \pm 1.72}$ |
| | GCN-LPA | $70.26 \pm 2.00$ | $26.70 \pm 1.47$ | $70.58 \pm 2.04$ | $33.65 \pm 2.61$ |
| 0.2 | Mlp | $68.37 \pm 0.55$ | $26.61 \pm 0.47$ | $54.47 \pm 0.19$ | $18.70 \pm 0.69$ |
| | DeepWalk | $\mathbf{86.22 \pm 0.39}$ | $\mathbf{47.80 \pm 0.31}$ | $\mathbf{84.15 \pm 0.56}$ | $\mathbf{48.70 \pm 0.51}$ |
| | LANC | $70.84 \pm 1.75$ | $27.62 \pm 0.74$ | $74.42 \pm 2.02$ | $34.24 \pm 1.42$ |
| | Gcn | $67.42 \pm 1.07$ | $25.31 \pm 1.09$ | $77.47 \pm 0.87$ | $31.59 \pm 0.63$ |
| | Gat | $67.07 \pm 1.07$ | $25.03 \pm 1.16$ | $64.55 \pm 0.60$ | $33.90 \pm 0.30$ |
| | GraphSage | $70.14 \pm 0.60$ | $27.99 \pm 1.07$ | $69.44 \pm 1.09$ | $32.84 \pm 0.79$ |
| | H2Gcn | $\underline{73.21 \pm 1.07}$ | $\underline{28.85 \pm 1.23}$ | $\underline{81.78 \pm 1.24}$ | $\underline{40.12 \pm 4.24}$ |
| | GCN-LPA | $69.67 \pm 1.67$ | $26.14 \pm 1.53$ | $71.02 \pm 1.71$ | $33.33 \pm 1.31$ |
| 0.5 | Mlp | $68.83 \pm 0.78$ | $27.57 \pm 1.10$ | $61.63 \pm 0.74$ | $21.95 \pm 0.40$ |
| | DeepWalk | $\mathbf{86.22 \pm 0.39}$ | $\mathbf{47.80 \pm 0.31}$ | $\underline{84.15 \pm 0.56}$ | $\mathbf{48.70 \pm 0.51}$ |
| | LANC | $73.21 \pm 2.79$ | $30.02 \pm 2.01$ | $77.05 \pm 1.16$ | $36.45 \pm 1.30$ |
| | Gcn | $67.55 \pm 1.08$ | $25.39 \pm 1.08$ | $77.25 \pm 0.85$ | $31.14 \pm 0.61$ |
| | Gat | $67.76 \pm 1.75$ | $25.49 \pm 1.46$ | $64.40 \pm 1.64$ | $32.92 \pm 0.88$ |
| | GraphSage | $74.70 \pm 0.49$ | $31.25 \pm 0.89$ | $76.24 \pm 0.47$ | $37.65 \pm 0.65$ |
| | H2Gcn | $\underline{76.16 \pm 0.47}$ | $\underline{32.85 \pm 0.28}$ | $\mathbf{84.96 \pm 0.55}$ | $\underline{42.70 \pm 0.27}$ |
| | GCN-LPA | $72.11 \pm 1.89$ | $27.80 \pm 0.77$ | $74.39 \pm 1.26$ | $36.84 \pm 1.02$ |
| 0.8 | Mlp | $70.17 \pm 0.81$ | $29.37 \pm 0.99$ | $69.53 \pm 0.46$ | $27.65 \pm 0.77$ |
| | DeepWalk | $\mathbf{86.22 \pm 0.39}$ | $\mathbf{47.80 \pm 0.31}$ | $\underline{84.15 \pm 0.56}$ | $\mathbf{48.70 \pm 0.51}$ |
| | LANC | $72.67 \pm 3.56$ | $29.67 \pm 3.13$ | $74.73 \pm 1.07$ | $35.32 \pm 2.06$ |
| | Gcn | $69.56 \pm 1.14$ | $25.79 \pm 1.14$ | $77.03 \pm 0.99$ | $30.10 \pm 0.71$ |
| | Gat | $67.93 \pm 0.75$ | $25.40 \pm 1.17$ | $66.41 \pm 3.55$ | $33.78 \pm 0.96$ |
| | GraphSage | $75.66 \pm 0.51$ | $32.39 \pm 1.50$ | $78.31 \pm 0.49$ | $39.25 \pm 0.86$ |
| | H2Gcn | $\underline{78.68 \pm 0.38}$ | $\underline{36.17 \pm 0.56}$ | $\mathbf{86.01 \pm 0.64}$ | $\underline{44.21 \pm 0.31}$ |
| | GCN-LPA | $72.62 \pm 1.46$ | $27.74 \pm 0.67$ | $73.99 \pm 1.06$ | $36.28 \pm 1.57$ |
| 1.0 | Mlp | $72.90 \pm 0.27$ | $31.52 \pm 0.52$ | $75.23 \pm 0.63$ | $34.29 \pm 0.07$ |
| | DeepWalk | $\mathbf{86.22 \pm 0.39}$ | $\mathbf{47.80 \pm 0.31}$ | $\underline{84.15 \pm 0.56}$ | $\mathbf{48.70 \pm 0.51}$ |
| | LANC | $74.13 \pm 1.61$ | $30.38 \pm 1.29$ | $75.26 \pm 0.93$ | $37.47 \pm 0.27$ |
| | Gcn | $70.74 \pm 0.80$ | $26.57 \pm 0.97$ | $79.22 \pm 0.82$ | $33.70 \pm 0.79$ |
| | Gat | $70.22 \pm 1.27$ | $26.35 \pm 0.93$ | $66.52 \pm 1.03$ | $36.01 \pm 0.72$ |
| | GraphSage | $78.71 \pm 0.59$ | $35.77 \pm 1.40$ | $80.77 \pm 0.17$ | $43.05 \pm 0.22$ |
| | H2Gcn | $\underline{79.97 \pm 0.19}$ | $\underline{38.73 \pm 0.91}$ | $\mathbf{87.09 \pm 0.12}$ | $\underline{46.71 \pm 0.37}$ |
| | GCN-LPA | $76.28 \pm 1.27$ | $31.91 \pm 1.07$ | $76.38 \pm 0.36$ | $39.10 \pm 2.03$ |

### A.5 Challenge of the Evaluation Metrics

The Area under the ROC curve (AUC) and the Area under the Precision-Recall curve (AUPR) are two widely accepted non-parametric measurement scores used by existing works. Nevertheless, as discussed in (Yang et al., 2015), the AUC score is sometimes misleading for highly imbalanced datasets like those in our work. In addition, AUPR might lead to over-estimation of the models' performance when the number of thresholds (or unique prediction values) is limited (Dong & Khosla, 2020). For such reasons, as suggested in (Dong & Khosla, 2020), we instead use the Average Precision (AP) score as our primary evaluation metric. Following

Table 11: Multi-label Node Classification results on Synthetic dataset with varying label homophily. All results are the mean of three random splits. The label homophiles (rounded up) are $[0.2, 0.4, 0.6, 0.8, 1.0]$

| **Label Homophily** | **Method** | Micro-F1 | Macro-F1 | AUC-ROC | AP |
|---|---|---|---|---|---|
| 0.2 | Mlp | $72.90 \pm 0.27$ | $\underline{31.52 \pm 0.52}$ | $\mathbf{75.23 \pm 0.63}$ | $\mathbf{34.29 \pm 0.07}$ |
| | DeepWalk | $66.62 \pm 0.62$ | $26.02 \pm 0.55$ | $52.19 \pm 0.91$ | $18.07 \pm 0.71$ |
| | LANC | $67.05 \pm 0.92$ | $25.12 \pm 2.20$ | $54.49 \pm 0.53$ | $18.95 \pm 0.43$ |
| | Gcn | $67.28 \pm 0.67$ | $25.21 \pm 1.04$ | $66.59 \pm 0.58$ | $26.06 \pm 1.07$ |
| | Gat | $65.09 \pm 0.89$ | $24.61 \pm 1.41$ | $51.43 \pm 0.33$ | $17.05 \pm 0.36$ |
| | GraphSage | $\underline{73.57 \pm 0.20}$ | $31.22 \pm 0.65$ | $71.72 \pm 0.30$ | $28.94 \pm 1.09$ |
| | H2Gcn | $\mathbf{73.80 \pm 0.59}$ | $\mathbf{31.86 \pm 0.89}$ | $\underline{73.24 \pm 0.08}$ | $\underline{29.69 \pm 0.53}$ |
| | GCN-LPA | $67.02 \pm 0.89$ | $25.24 \pm 1.31$ | $49.92 \pm 0.93$ | $17.02 \pm 0.31$ |
| 0.4 | Mlp | $72.90 \pm 0.27$ | $31.52 \pm 0.52$ | $75.23 \pm 0.63$ | $34.29 \pm 0.07$ |
| | DeepWalk | $\mathbf{88.79 \pm 0.20}$ | $\mathbf{54.74 \pm 1.73}$ | $\underline{85.74 \pm 0.33}$ | $\mathbf{52.15 \pm 1.41}$ |
| | LANC | $75.16 \pm 0.77$ | $33.69 \pm 0.94$ | $76.53 \pm 0.86$ | $37.99 \pm 0.74$ |
| | Gcn | $72.16 \pm 1.70$ | $26.95 \pm 1.18$ | $80.19 \pm 0.75$ | $34.29 \pm 0.88$ |
| | Gat | $71.50 \pm 1.30$ | $27.58 \pm 1.11$ | $69.96 \pm 1.69$ | $35.86 \pm 0.59$ |
| | GraphSage | $79.09 \pm 0.33$ | $36.05 \pm 1.56$ | $81.00 \pm 0.35$ | $42.57 \pm 0.49$ |
| | H2Gcn | $\underline{81.30 \pm 0.25}$ | $\underline{40.54 \pm 0.74}$ | $\mathbf{87.91 \pm 0.04}$ | $\underline{48.36 \pm 0.44}$ |
| | GCN-LPA | $78.11 \pm 1.38$ | $32.71 \pm 1.47$ | $78.00 \pm 0.70$ | $40.80 \pm 0.20$ |
| 0.6 | Mlp | $72.90 \pm 0.27$ | $31.52 \pm 0.52$ | $75.23 \pm 0.63$ | $34.29 \pm 0.07$ |
| | DeepWalk | $\mathbf{95.94 \pm 0.05}$ | $\mathbf{82.58 \pm 0.19}$ | $\mathbf{95.32 \pm 0.80}$ | $\mathbf{81.34 \pm 0.95}$ |
| | LANC | $80.36 \pm 1.81$ | $39.65 \pm 3.14$ | $81.85 \pm 2.66$ | $43.42 \pm 3.61$ |
| | Gcn | $75.47 \pm 0.46$ | $29.43 \pm 0.43$ | $84.08 \pm 0.72$ | $38.78 \pm 0.33$ |
| | Gat | $75.13 \pm 0.39$ | $29.26 \pm 0.56$ | $72.85 \pm 0.79$ | $39.01 \pm 0.23$ |
| | GraphSage | $82.46 \pm 0.67$ | $40.46 \pm 1.67$ | $83.24 \pm 0.89$ | $45.79 \pm 0.91$ |
| | H2Gcn | $83.40 \pm 1.94$ | $44.32 \pm 1.59$ | $\underline{89.98 \pm 0.86}$ | $\underline{51.19 \pm 1.92}$ |
| | GCN-LPA | $\underline{85.45 \pm 2.42}$ | $\underline{47.67 \pm 4.72}$ | $83.96 \pm 2.22$ | $49.54 \pm 3.54$ |
| 0.8 | Mlp | $72.90 \pm 0.27$ | $31.52 \pm 0.52$ | $75.23 \pm 0.63$ | $34.29 \pm 0.07$ |
| | DeepWalk | $\mathbf{96.53 \pm 0.61}$ | $\mathbf{89.25 \pm 1.96}$ | $\mathbf{95.81 \pm 0.47}$ | $\mathbf{86.93 \pm 1.92}$ |
| | LANC | $79.35 \pm 0.97$ | $46.03 \pm 2.24$ | $87.75 \pm 0.60$ | $48.10 \pm 1.17$ |
| | Gcn | $80.72 \pm 0.55$ | $36.60 \pm 1.31$ | $86.82 \pm 0.62$ | $44.98 \pm 0.40$ |
| | Gat | $77.35 \pm 0.49$ | $31.63 \pm 1.10$ | $83.10 \pm 0.69$ | $42.82 \pm 0.86$ |
| | GraphSage | $84.22 \pm 0.38$ | $48.16 \pm 1.42$ | $87.37 \pm 0.34$ | $53.34 \pm 0.84$ |
| | H2Gcn | $86.00 \pm 2.63$ | $55.85 \pm 4.92$ | $\underline{91.61 \pm 1.64}$ | $57.17 \pm 2.57$ |
| | GCN-LPA | $\underline{88.96 \pm 0.68}$ | $\underline{64.09 \pm 4.18}$ | $89.53 \pm 1.07$ | $\underline{60.44 \pm 4.41}$ |
| 1.0 | Mlp | $72.90 \pm 0.27$ | $31.52 \pm 0.52$ | $75.23 \pm 0.63$ | $34.29 \pm 0.07$ |
| | DeepWalk | $80.25 \pm 0.11$ | $\underline{62.80 \pm 0.46}$ | $83.83 \pm 1.08$ | $55.16 \pm 0.67$ |
| | LANC | $83.37 \pm 0.96$ | $60.77 \pm 3.51$ | $\underline{92.53 \pm 1.27}$ | $\underline{62.85 \pm 3.01}$ |
| | Gcn | $81.61 \pm 0.25$ | $42.19 \pm 0.45$ | $86.48 \pm 0.31$ | $49.32 \pm 0.92$ |
| | Gat | $79.37 \pm 0.64$ | $34.67 \pm 1.87$ | $87.20 \pm 2.15$ | $43.90 \pm 2.90$ |
| | GraphSage | $82.15 \pm 0.49$ | $46.06 \pm 0.99$ | $89.73 \pm 0.45$ | $55.25 \pm 0.42$ |
| | H2Gcn | $\mathbf{91.59 \pm 1.74}$ | $\mathbf{76.09 \pm 5.96}$ | $\mathbf{93.94 \pm 1.03}$ | $\mathbf{65.20 \pm 5.71}$ |
| | GCN-LPA | $\underline{86.92 \pm 1.13}$ | $60.71 \pm 3.01$ | $90.75 \pm 0.46$ | $58.28 \pm 3.02$ |

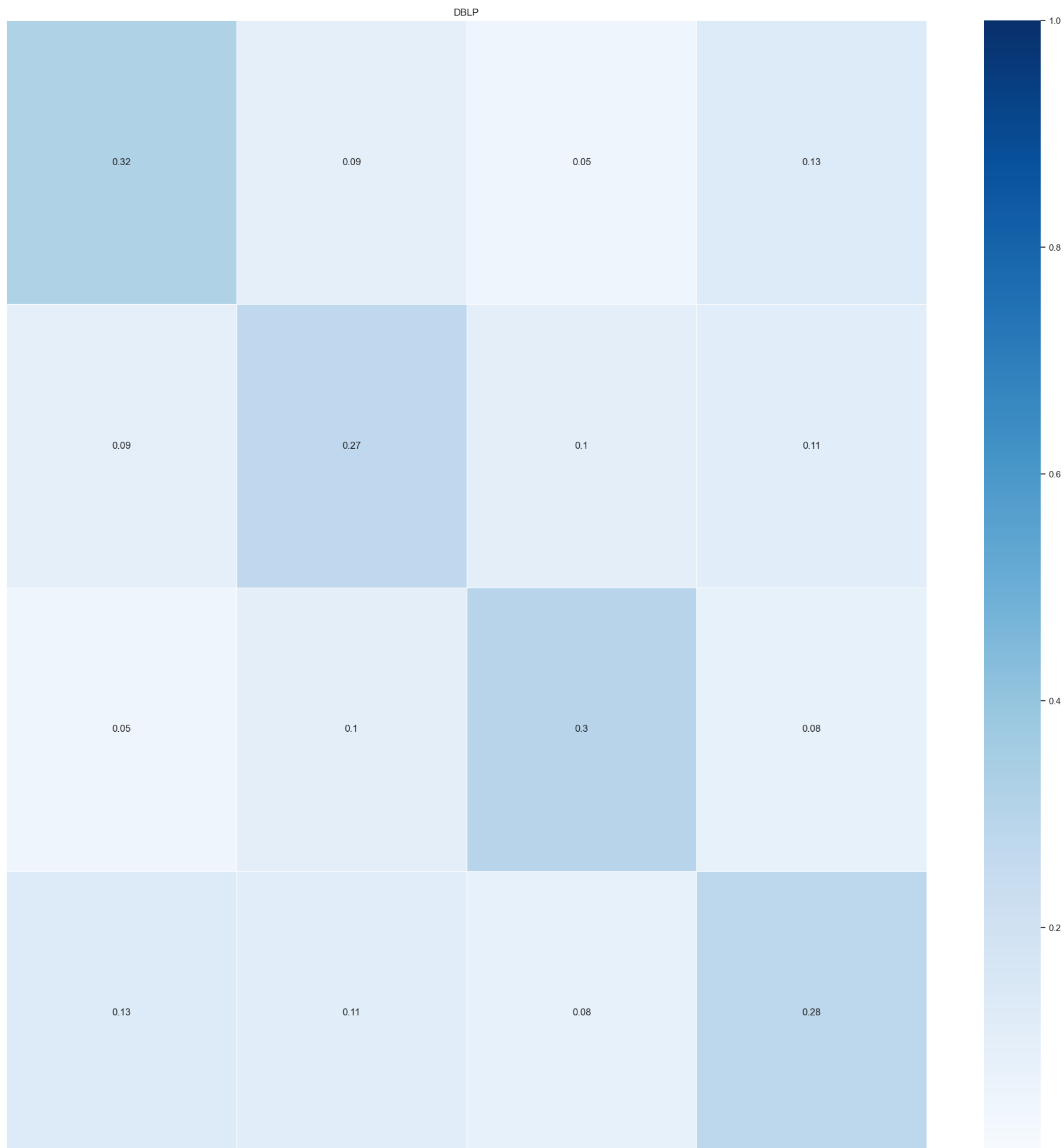

Figure 7: Cross class Neighborhood Similarity in DBLP

previous works, we also report the F1 score. As it is a threshold-based metric we emphasize that it might be biased when the benchmarked models have different prediction score ranges.

### A.6 Cross-class Neighborhood Similarity Plots

In this section, we put the heat maps of the cross-class neighborhood similarity for all the datasets used in this work. We use color coding, where a darker shade in the cell indicates a stronger cross-class neighborhood similarity.

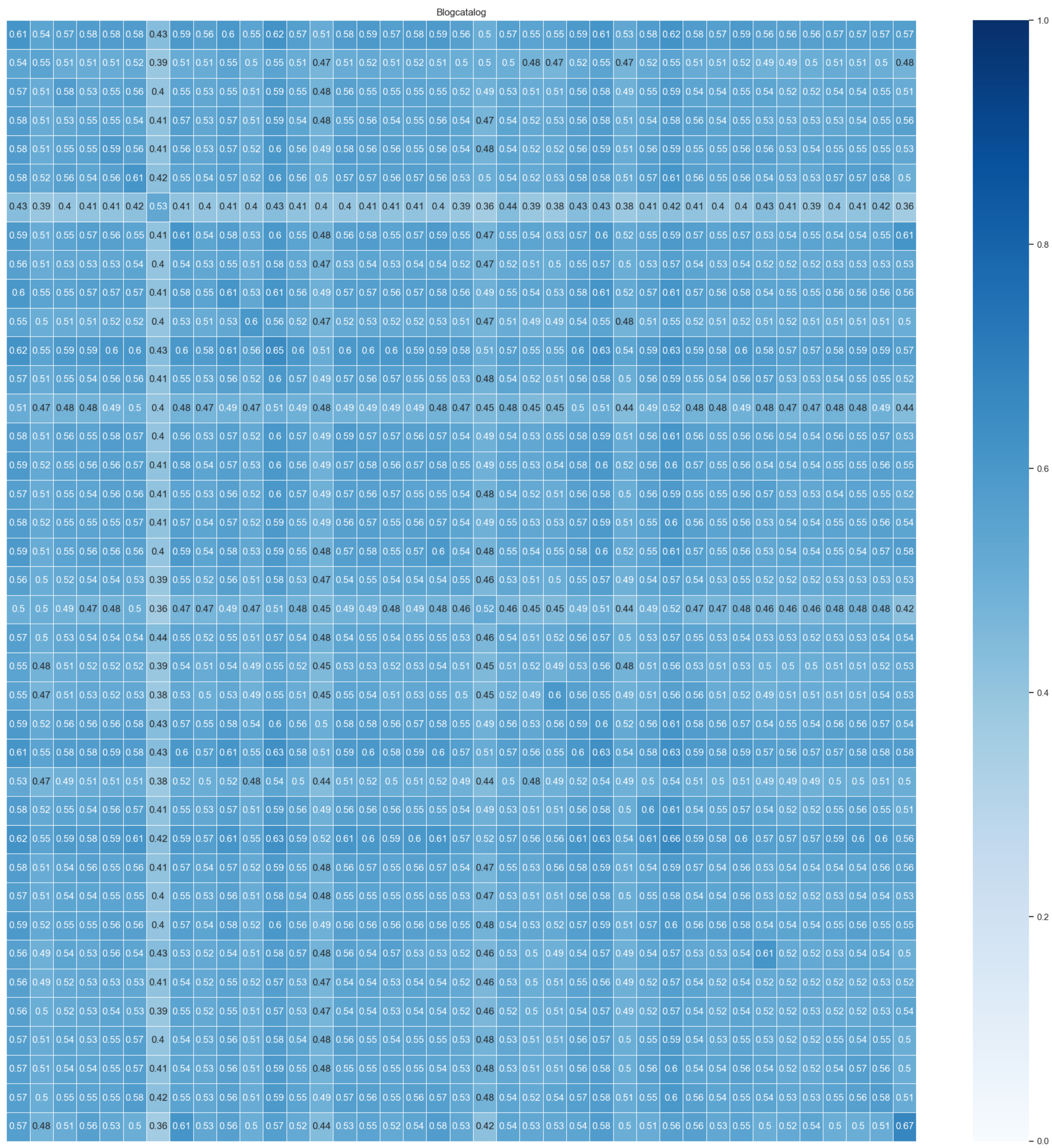

Figure 8: Cross class Neighborhood Similarity in BlogCat

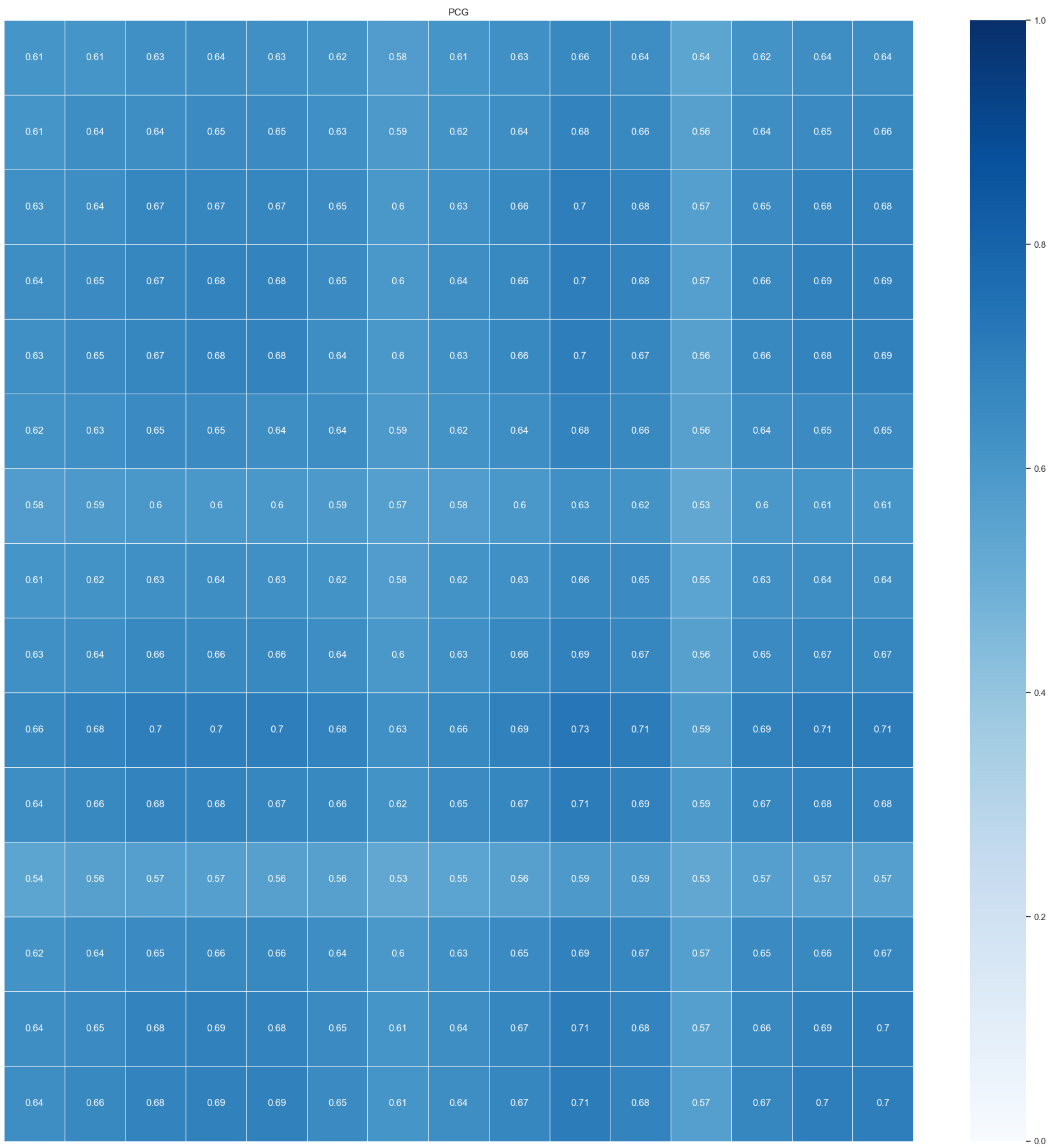

Figure 9: Cross class Neighborhood Similarity in PCG

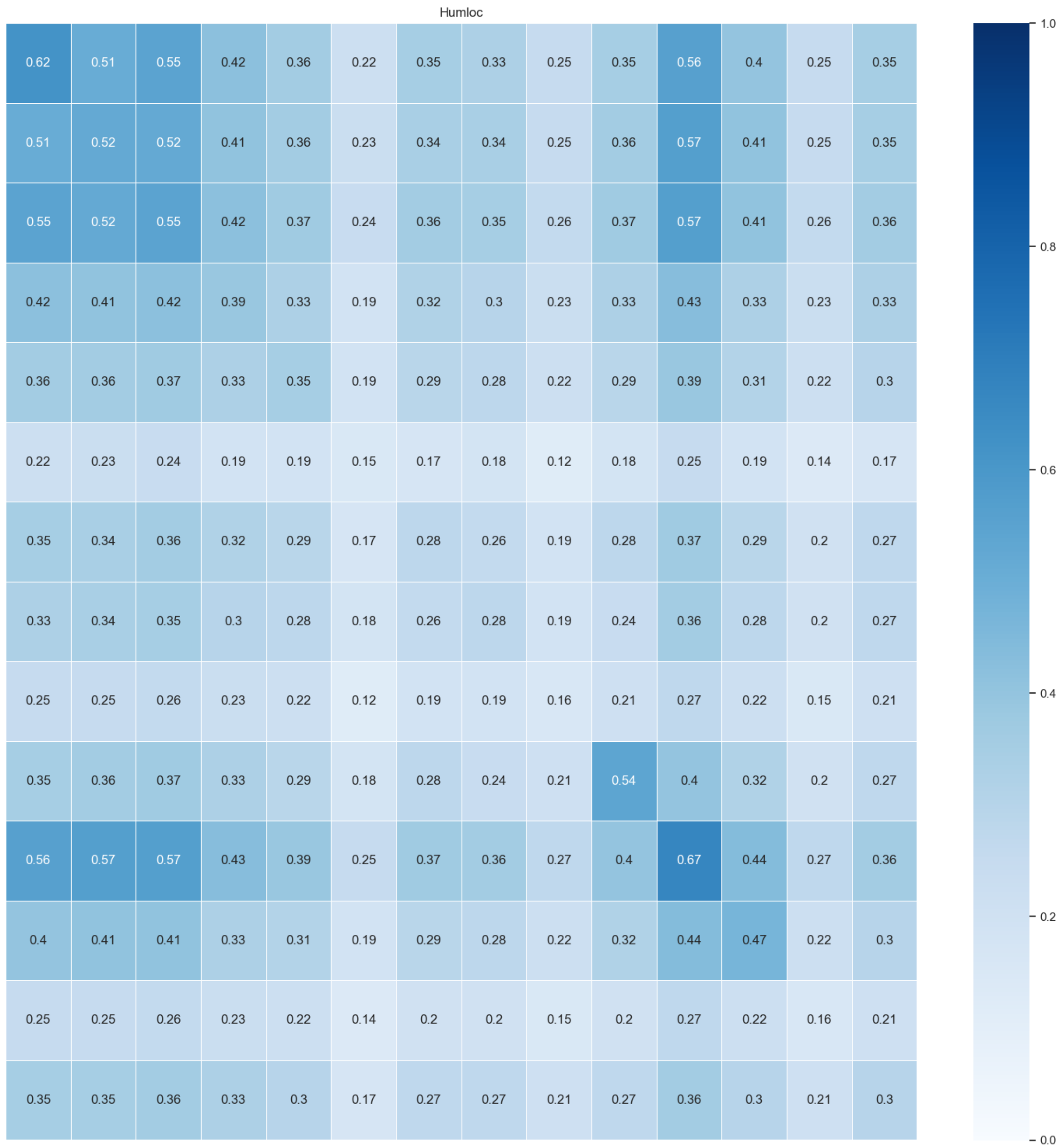

Figure 10: Cross class Neighborhood Similarity in HumLoc

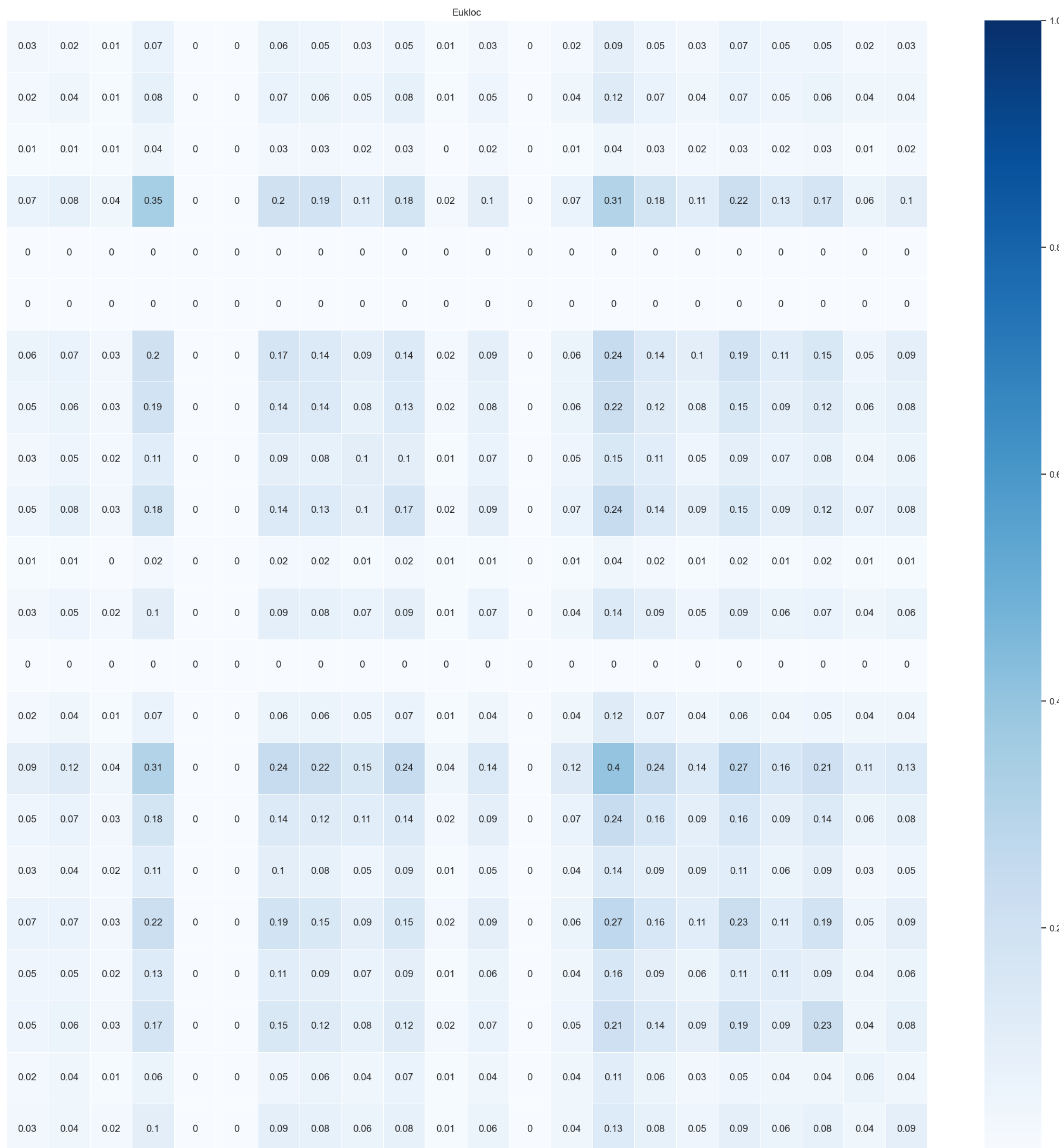

Figure 11: Cross class Neighborhood Similarity in EukLoc

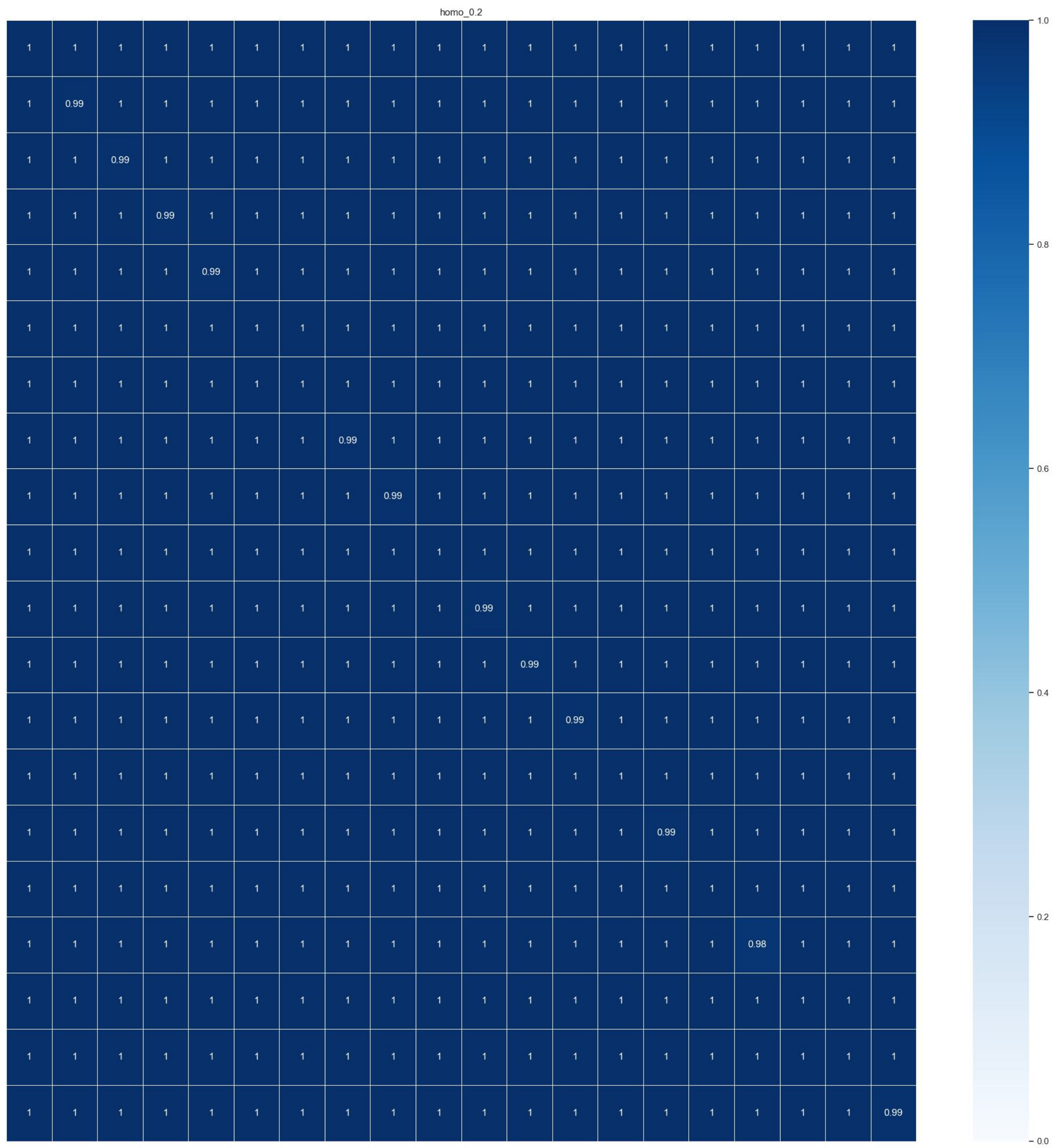

Figure 12: Cross-class Neighborhood Similarity in graph label homophily=0.2

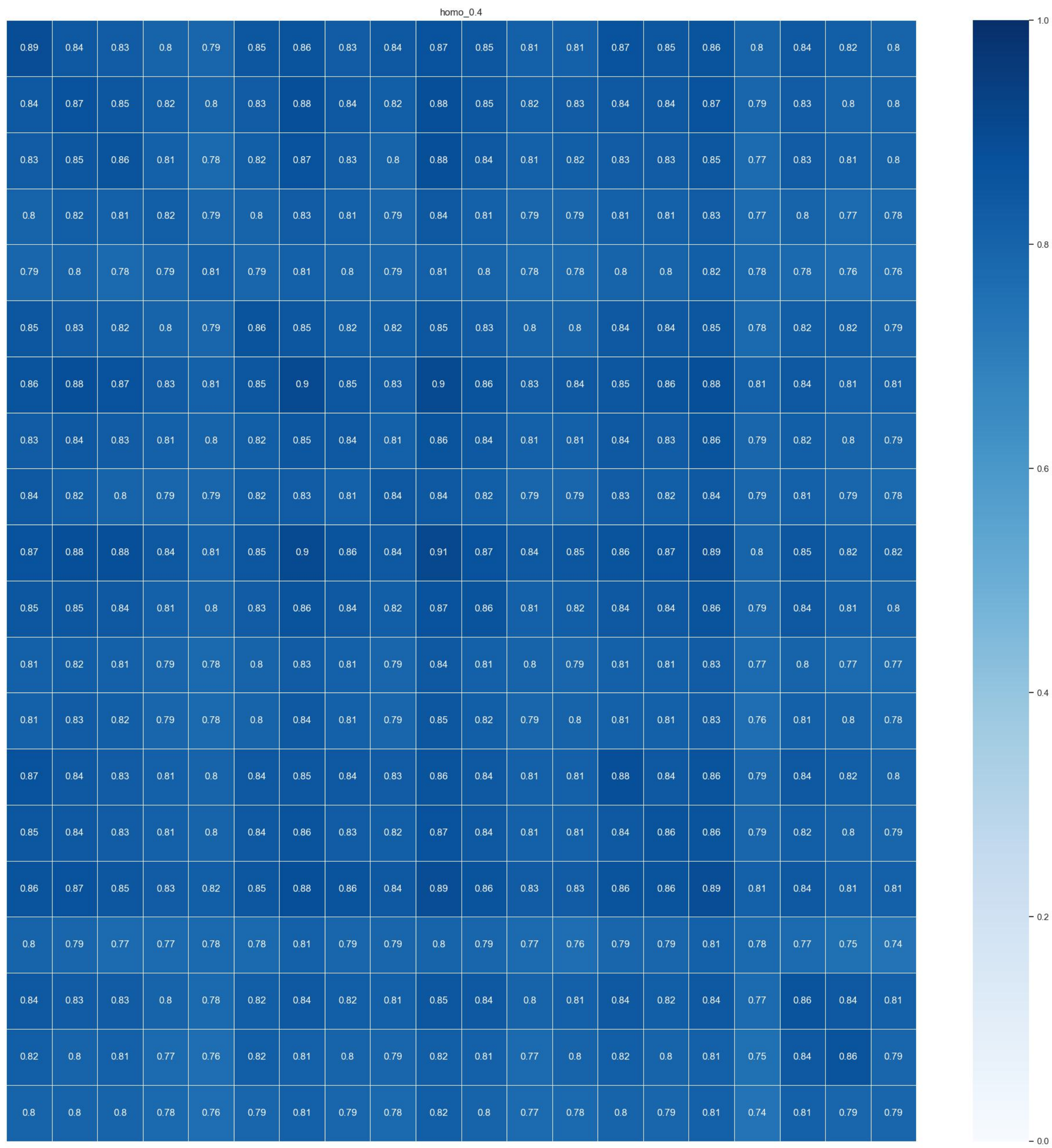

Figure 13: Cross-class Neighborhood Similarity in graph label homophily=0.4

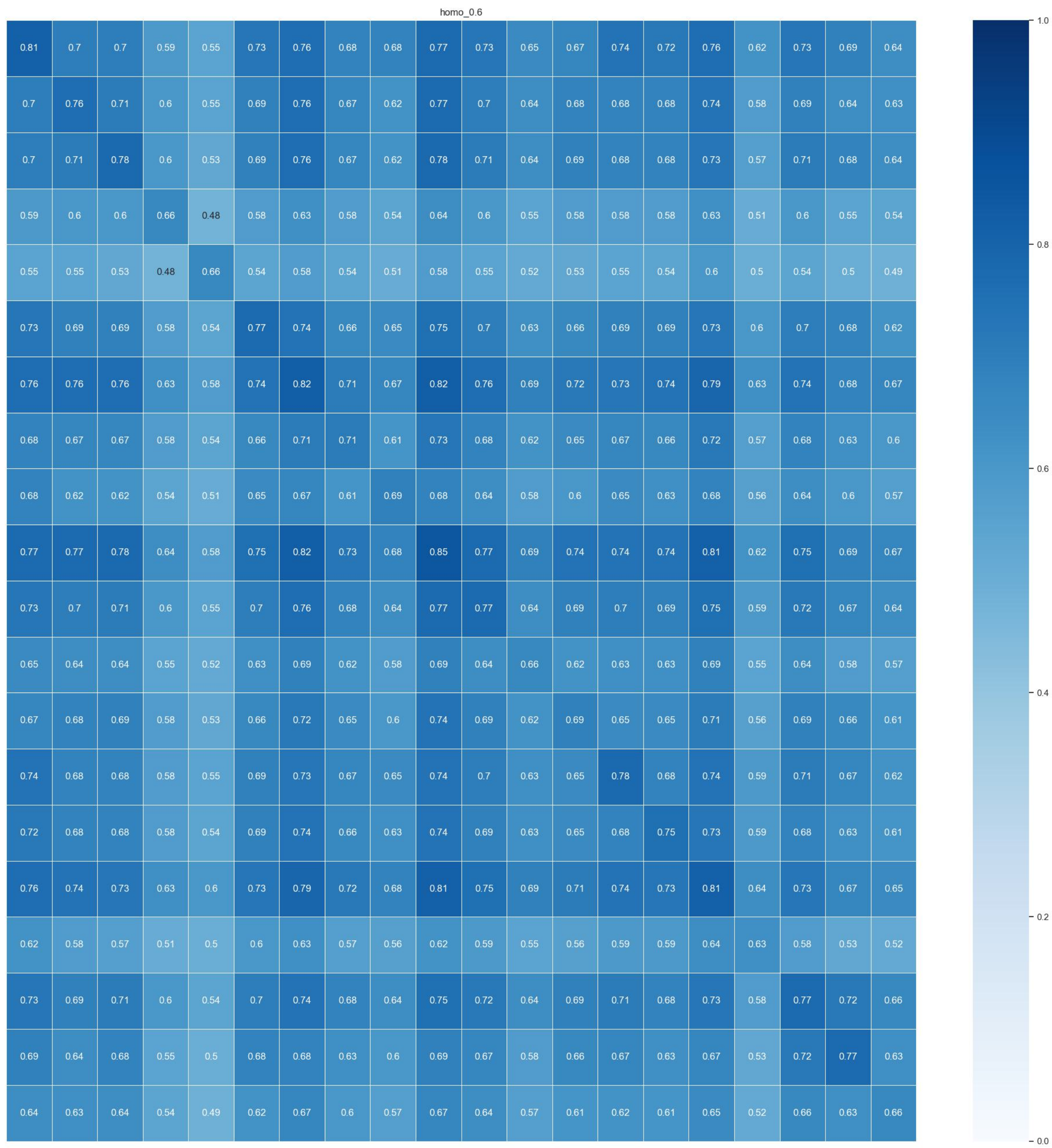

Figure 14: Cross-class Neighborhood Similarity in graph label homophily=0.6

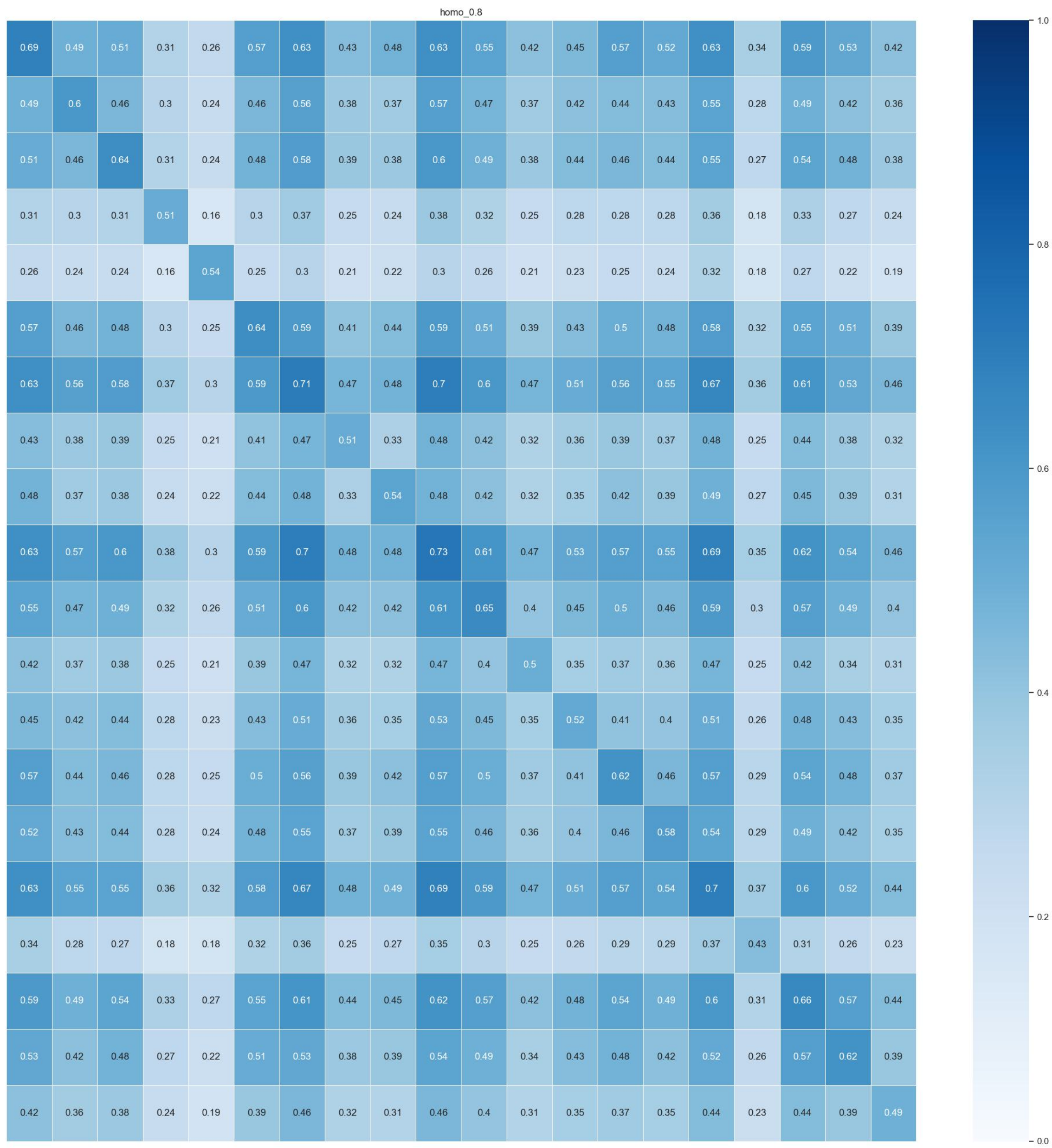

Figure 15: Cross-class Neighborhood Similarity in graph label homophily=0.8

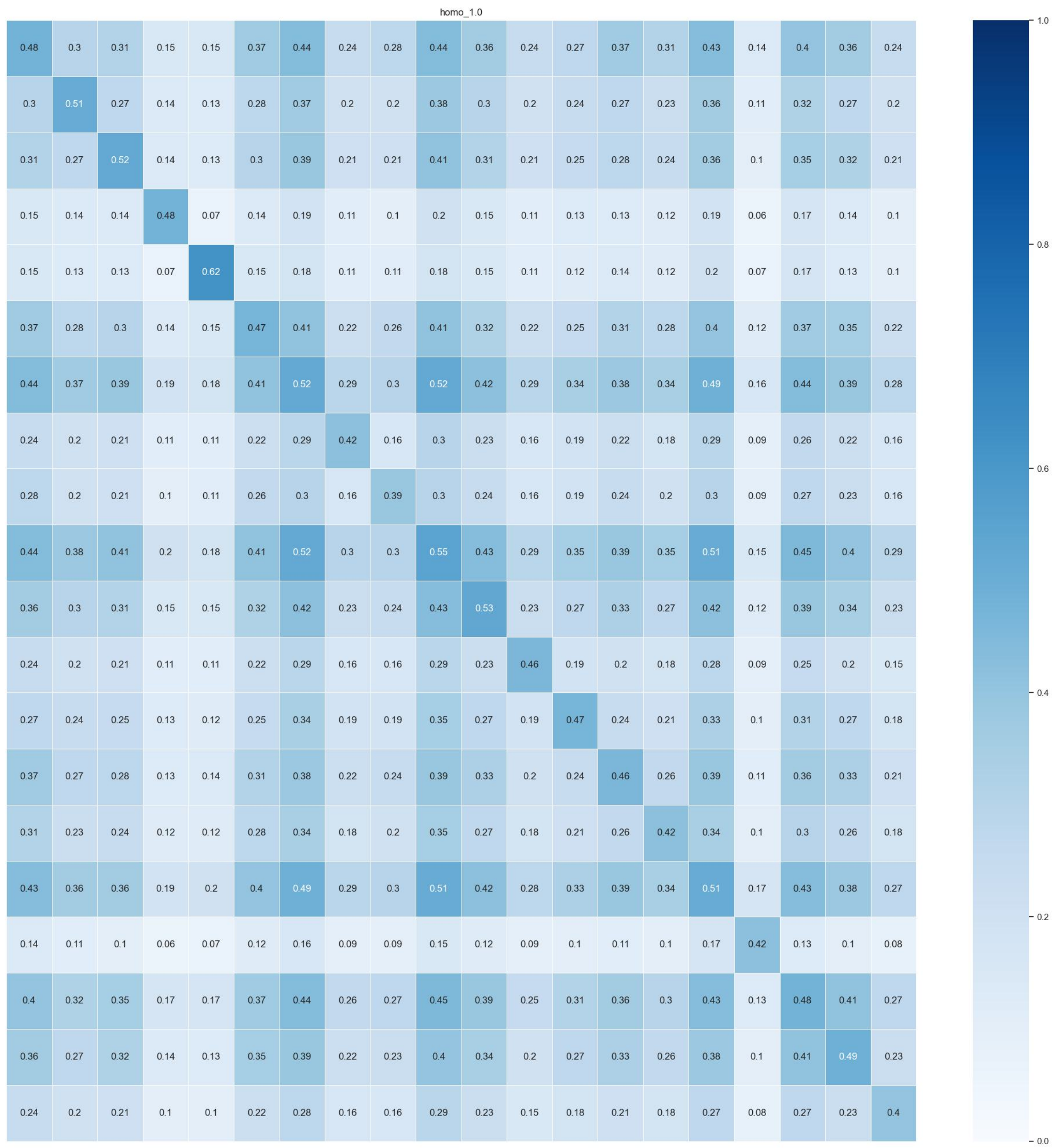

Figure 16: Cross-class Neighborhood Similarity in graph label homophily=1.0