# OpenReview forum: "Multi-label Node Classification On Graph-Structured Data"
_TMLR — Accepted by TMLR_

### Review · Reviewer_7Tut · 2023-03-08

**Summary Of Contributions:**

In this paper, the authors make several distinct contribution on the topic of multi-label node classification in graphs. They first revisit the notions of homophily and neighborhood similarity, which are generally associated with the success (or failure) of GNNs. They analyze some existing datasets under this light and comment on some discutable choices made in the literature. Then, they propose three new curated datasets of biological data for multi-label node classification. Next, they propose a simple random model for generating graphs with multi-labelled nodes, with some adjustable parameters. Finally, they propose a novel variant of GNNs that use both features and labels as input, with some attentional mechanism between the two, and benchmark several baselines and models on multi-label tasks.

**Audience:**

Yes

**Broader Impact Concerns:**

N/A. New datasets are provided, but new data is not collected, only curated. Sufficient reference is given.

**Claims And Evidence:**

Yes

**Requested Changes:**

See above. The paper has merits, but suffers from a lack of rigorousness and completeness that would greatly improve the reading.

**Strengths And Weaknesses:**

Strength:
- the paper is generally quite well-written and clear
- the paper is very complete and explore its topic quite thoroughly
- the authors propose new interesting insights on graphs with multi-labelled nodes, and the use of node features and node labels

Weaknesses:
- there is a somewhat lack of rigorousness and/or completeness on several notions and definitions introduced by the authors. Especially on the (few) mathematical formulations: objects are generally vaguely defined, no dimensionality is given, many notions and notations are not defined and can be ambiguous, etc.
- For instance, the very definition of multi-label is not given: is it that, among a set of labels, a node can have several, or is it that each node is associated to a multidimensional label vector, with several number of classes for each coordinate? (I deduced that it is the first one, but it took me some back and forth reading). In this light, Def. 2 is quite not clear (what does $i \in V_c$ means in a multi-label context?).
- As another example, the authors should mention from the beginning which task they are adressing: I'm guessing semi-supervised learning? Transductive? Inductive? The training of the the proposed GNN is unclear: why is it a good strategy to consider purely an unsupervised graph reconstruction loss? If no label are available (inductive settings), what it the advantage of doing label propagation?
- Similarly, in the random model, it took me some times to understand that the $x_i$ were the continuous points in the hypersphere, playing the role of node features, and not the labels, since $d(x_i,x_j)$ refers to a distance between labels! By the way, in the many existing Latent Position Models (LPM) of random graphs, it is much much more classical to consider a true distance between the $x_i$ and not the labels. Some discussion with the literature is needed here.
- the authors mention several times the problem with the AUROC approach, but do not give true mathematical justification. Some more details would be nice.

---

> ### Author Response · Authors · 2023-04-30
> **Answer to Reviewer 7Tut**
>
> We thank the reviewer for the suggestions. We have updated the paper with the suggestions. We marked the changes in blue to make the review easier. We welcome any further constructive discussion.
>
>
> **1.Notations**: Thanks for your comments. we updated the corresponding notations and provided explicit definitions wherever required in the paper. Please check our paper for updates.
> ___
> **2.Problem Setting**:
>
>  *   The definition of the multi-label dataset is already given in section 2.1 under the Problem Setting. In the multi-label node classification task, a node  is associated with multiple labels selected from a predefined set of labels. In addition, we define $\mathcal{V}_{c}$ as the set of nodes with one of their labels as label $c$ And $i$ belongs to $ \mathcal{V} _{c}$ means $i$ is one of the nodes in set $ \mathcal{V} _{c}$ .
>
>  *   The task setting was stated in the section 2.1 problem setting by stating” We deal with the transductive setting multi-label node classification problem, where the features and graph structure of the test nodes are present during training.”
> The reason of the choice of unsupervised learning loss is given in section 5 “Intermediate predicting and model training”. To summarize, solely relying on supervised loss and using the true labels of the training nodes as input for the label propagation phase can lead to overfitting and hinder the model's generalization to unseen data. To mitigate this issue, we introduced an unsupervised graph re-construction loss.
>
> *   Furthermore, inductive learning setting means that the model is trained and tested on different graphs but can still be trained with the true labels of the training data as part of the input. While in our current paper we solely focus on transductive learning , our model is adaptable to the inductive learning setting. A thorough investigation of our model in inductive settings will be considered in future work.
>
> ___
> **3.Choice of label distance**  as in d(xi,xj) to compute edge probabilities in the synthetic graph model : We used label vector instead of feature vector: as already described in the text on page 8 section 4, we define label homophily as the similarity between the labels of connected nodes in the graph. We use distance between labels and not between features (coordinates) to construct datasets based on label homophily. Because the intention to propose the graph generator model is to have synthetic graphs with varying label homophily and study the impact of varying homophily on the performances of the models from various categories. We updated the notations to avoid any ambiguity.
> ___
> **4.Justification of problem in AUROC**:
>
> Note that AUROC measures the area under the area under ROC curve. The ROC curve plots true positive rate (TPR) against the false positive rate (FPR)).  For multi-label classification, we compute AUC-ROC score for each class, i.e., if the total number of labels/classes is C then we solve C binary classification problems. Corresponding to each test node $u$ and class/label $c$, the model outputs a score $p(c)$ such that $p(c)$ is the (predicted) probability that the $u$ has a label $c$.  So, if test nodes have missing labels, the model which predicts very low scores for all $c$ is doing the right thing, it is predicting the corresponding negative class. In other words, there will be a high imbalance in the size of positive and negative class for each binary classifier.  AUROC is known to be insensitive to such class imbalance (as also mentioned in [1] and discussed at length in here: https://towardsdatascience.com/precision-recall-curve-is-more-informative-than-roc-in-imbalanced-data-4c95250242f6
>
>  And produce overly optimistic results due to the simple fact that it does not consider the precision (of positive class) as done in Average Precision or AURP scores. In the following we also illustrate this issue with an example:
> For instances, consider the case of a dataset which has 10 positives and 100,000 negatives. We have two model A and B, A predicts 900 positives, in which 9 of them are true positives, and B predicts 90 positives, in which 9 of them are true positives. Intuitively, model B should be rated with a higher performance score because it does not output as many false positives. However, the TPR and FPR calculated for the two models are:
>
> Model A: TPR = 9/10 = 0.9 and FPR = (900–9)/100,000 = 0.00891
> Model B: TPR = 9/10 = 0.9 and FPR = (90–9)/100,000 = 0.00081
>
> TPR is, as expected, is exactly the same between both models. On the other hand, since the number of negatives largely dominates that of positives, the difference of FPR between both models (0.00891–0.00081 = 0.0081) is lost in the sense that it can be rounded to almost 0. In other words, a large change in the number of false positives resulted in a tiny change in the FPR due to large size of negative class.

---

> > ### Author Response · Authors · 2023-04-30
> > **Reference mentioned in the answers to Reviewer 7Tut**
> >
> > **Reference**
> >
> >  [1] Davis, Jesse, and Mark Goadrich. "The relationship between Precision-Recall and ROC curves." Proceedings of the 23rd international conference on Machine learning. 2006.

---

### Review · Reviewer_WHFe · 2023-04-11

**Summary Of Contributions:**

This paper studies multi-label node classification on graphs. It starts with analyzing existing datasets by computing the label homophily and the cross-class neighborhood similarity. The authors introduce three new biological interaction datasets as well as a multi-label graph generator to synthesize models with tunable parameters. In section 5, they introduce a variant of GNN called layerwise feature label fusion (LFLF), where both features and labels are propagated through GNN layers and then aggregated with an attention-type layer. The learning task is a link prediction task. Finally, experiments are conducted on real datasets and synthetic ones where LFLF is shown to be optimal.


**Audience:**

Yes

**Broader Impact Concerns:**

I do not think there is a need for a BIS.

**Claims And Evidence:**

No

**Requested Changes:**

Given the major weaknesses discussed above, I am recommending rejecting the paper.
Here are some other required changes:
- overall, the results obtained by DeepWalk are very strong, but there is a more advanced version Node2Vec, and comparisons should be done with Node2Vec
- in section 3, the authors claim that OGB-Protein is not suitable for experiments because a lot of labels are missing. But then in Section 7, they use this dataset to show that their algorithm is better than others. How did they deal with the missing labels?
- Figures 1,3, and 5a are histograms; why do the authors use these non-standard plots?
- Figures for Cross class neighborhood similarity are not readable.
- on page 5, I do not understand the following: "Specifically, each node pair(i,j)would contribute a total of 1 units to CCNS similarity over all possible class pairs."
- on page 6, I do not understand the following: "We in fact observed that increasing the number of training epochs (which encourage the model to decrease training loss by predicting the negative class)increased the AUROC score whereas other metrics like AP or F1 score dropped or stayed unchanged."
- on page 10, I do not understand the following: "We comment that in our current experimental setup, to save extra computational time, we set Lk=L0 for all k but always update Lk."

**Strengths And Weaknesses:**

Strengths:
1- Section 3 on the analysis of the existing dataset and the new datasets is an interesting contribution. The main point made by the authors is that the success of GNN is usually argued in terms of label homophily, but the empirical label homophily computed for existing datasets is pretty low.
2- In order to understand better the possible reasons for the success of GNN, section 4 introduces a graph generator allowing the authors to tune various parameters like clustering and label homophily. This is a very interesting approach.

Weaknesses:
1- I have a general problem with the message of this paper. On the one hand, the authors show that the label homophily and the cross-class neighborhood similarity are low in real datasets, and GNN layers should take these facts into account. On the other hand, the authors propose a graph generator with a possible high level of homophily and show that on such synthetic data, their GNN layer (LFLF) is performing better. That's fine, but then what are the practical implications? I understand that LFLF has been designed to work better when homophily is high, but this is not the case in most real-world datasets. Results in Table 3 about LFLF for real datasets are then quite surprising and not explained.
2- I find it problematic that the analysis of datasets is completely ignoring the features of the nodes. Indeed both definitions 1 and 2 only take into account the graph structure and the labels. Features on nodes are not taken into account. I do not understand why and the authors should explain this choice.
3- The general writing of this paper should be improved. There are a lot of crucial places where the paper does not define important concepts:
a- in definition 2, define d(i) the histogram properly and define cosine similarity.
b- on page 8, the description of the multi-label generator is completely unclear to me.
c- LFLF defined in section 5 is unclear. What is the label correlation matrix? What is the negative sampling distribution? What changes in your model variants?
d- in your experiments, you should be more precise. I do not understand how you get labels from node embeddings. In Section 5, your GNN learns embedding for nodes; how do you get a classifier from there? Similar question with DeepWalk? For the GCN, GAT... what task did you train them on?
e- in section 7, how do you vary what you call the feature quality? What is the definition of feature quality? What is the definition of original versus irrelevant features?

---

> ### Author Response · Authors · 2023-04-30
> **Answers to the reviewer WHFe**
>
> We thank the reviewer for the suggestions. We have updated the paper with the suggestions. We marked the changes in blue to make the review easier. We welcome any further constructive discussion.
>
>
> **1.Main Message**: As mentioned in the introduction, the first crucial message of the paper is that “the multi-label datasets do not follow the black and white separation of homophily and heterophily” as in multi-class datasets. In a multi-class dataset with a low label homophily one can conclude that the node’s label information cannot be inferred alone from the label information of its immediate neighbours. This is NOT always the case for multi-label datasets with low homophily. A node’s label information in a multi-label dataset can be inferred alone from the labels of its one-hop neighbours even if it’s calculated label homophily is low. We illustrate that GNNs fail to exploit the neighbourhood's label information in case of multi-label datasets unless it has homophily of very high value close to 1, i.e., when all nodes in the neighbourhood have essentially the same label set. This is where our approach comes in. We provide a method to exactly take into account different levels of label similarity between neighbourhood nodes in a multi-label dataset.
>
> ___
> **2.Analysis on Table 3**: The analysis and comparison of the performances of LFLF with other baselines on real-world datasets are explicitly explained in section 7.1. We compared and analysed the performances of the models on each of the real-world datasets. As already mentioned in the previous answer, by effectively learning and combining feature and label embeddings, LFLF is exploiting the label structure of the neighborhood which despite of low label homophily is still informative of the node’s label set.
>
> ___
> **3.Feature ignorance in definition 1 and 2**: We specifically choose the similarities based on neighborhood labels as we wanted to show the ineffectiveness of GNNs in utilizing these similarities that are already from the training data in multi-label datasets. We also empirically show that when features are noisy exploiting these neighborhood label information becomes essential (see section 7.2.1). An in-depth study of interplay of features and neighbourhood label information is out of scope of this work.
>
> ___
> **4. Writing of the paper**:
>
> *   To maintain the readability of the paper, we didn’t initially explicitly define the well-known terms like histogram and cosine similarity. The definition of cosine similarity is added to the paper. We have added those definitions now.
> *   we updated the description of the graph generator model in the paper. If it is still unclear, we kindly request the reviewers to point out which exact part is unclear.
> *   Sorry for the confusion, we added more description in the section 5 “Label propagation”. And as described in section 5 “model variation”, our framework provides the users a plug and play scheme with customizable choices of GNN layers. In our work, we take GCN layer and GraphSAGE layer as an example, the users are free to choose other GNN layers as well.
> *   As already described in the paper, the learned embeddings of the training nodes are used as the input to train a logistic regression model to infer the label of the test nodes. The training setting for other baseline models are already given in the section 6 “Experiment” in the paragraph baselines. The synthetic dataset construction and definition of feature quality are given in the section 4.
> ___
> **5.Strong performance of DeepWalk and Node2Vec**: Note that the main message and conclusion of the paper will NOT change if DeepWalk is replaced by Node2Vec. In particular, in the current literature on GNNs, simpler node embedding techniques like DeepWalk are hardly compared. The main goal of adding DeepWalk in our benchmark was to bring it to the attention of the community that simpler baselines could be powerful and should not be missed. Moreover, previous works such as [1] already pointed out the strong performance of DeepWalk (as compared to Node2vec and several other embedding methods) on several datasets, which further justifies the choice of DeepWalk in the current work
>
> ___
> Reference
>
>
> [1] Megha Khosla, Vinay Setty, and Avishek Anand.  “A Comparative Study for Unsupervised Network Representation Learning.” IEEE Transactions on Knowledge and Data Engineering. 2019

---

> > ### Author Response · Authors · 2023-04-30
> > **Answers to Reviewer WHFe Part 2**
> >
> > ___
> > **6.About dataset OGB-Protein**: We emphasize that OGB-Protein can be used for experiments but NOT with AUROC as the evaluation metric. In short,  missing labels and the modelling of multi-label classification problem as independent binary classification tasks cause a high class imbalance.  AUROC is known to show exaggerated results in such cases as also explained in answer to the last question from Reviewer 7Tut. We did not perform any pre-processing on ogbn-proteins in our paper. Ogbn-proteins is a widely used benchmark dataset in many papers. However, it has not been analysed properly before. By putting the analysis of the datasets before the experiment section, we point out that for the datasets that have a high imbalance between positive and negative classes, AUROC cannot and should not be used for evaluation. The reason being that when modelling multi-label classification as multiple binary classification tasks, the missing label is inherently considered as a negative class, which leads to over-exaggeration of AUROC scores.
> > ___
> > **7.Plots**
> > *   Nonstandard plots: The plots are violin plots. A violin plot is a statistical graphic for comparing probability distributions. It is similar to a box plot, with the addition of a rotated kernel density plot on each side.  It is more informative than a plain box plot. While a box plot only shows summary statistics such as mean/median and interquartile ranges, the violin plot shows the full distribution of the data. The difference is particularly useful when the data distribution is multimodal (more than one peak). In this case a violin plot shows the presence of different peaks, their position and relative amplitude.
> > *   Unreadable: We already use colour coding for visualizing the CCNS matrices. the explanations are given in section 2. The cross-class neighbourhood similarity is showed by the colour shades of the grids in the plots. The darker a grid is, the higher the cross-class neighbourhood similarity is between the nodes from the corresponding class pair. For your convenience we also add the bigger plots in the appendix section A.6.
> > ___
> > **8.Definition of CCNS**: We have elaborated the definition to improve its understanding.
> > ___
> > **9.Clarification of the statement about AUROC**:Here we again want to stress on the lack of effectiveness of metric like AUROC for evaluation. As already explained in reply to Reviewer 7Tut, AUROC is insensitive to the imbalance in the class size. When the size of the negative class is much larger than the positive class in both train and test sets, a model when trained for a large number of epochs will overfit to the training data and produce low scores corresponding to all classes (see also reply to Reviewer 7Tut,  in which we elaborate the setting of using C binary classifiers, one for each class). AUROC will not affected as much as other metrics such as AP and F1. This is because of the fact that AUROC does not consider precision of the results whereas other metrics do. Considering precision of positive class becomes even more important in datasets with high class imbalance.
> > ___
> > **10.Label correlation matrix**: Sorry for the confusion, we updated the typo in the paper. Check section 5 for detailed description.

---

### Review · Reviewer_h2ys · 2023-04-17

**Summary Of Contributions:**

The paper studies the problem of multi-label node classification using GNN models. Specifically, it makes three different contributions.
*(Metrics)*. First, it proposes a new metric to measure label homophily in the case of multi-label graph datasets. Four real-world datasets are explored regarding their label properties, having implications for the usage of the AUROC metric.  *(Datasets)*. Second, the paper introduces new multi-label datasets as well as a generator to produce synthetic graphs.  *(Model)*. Finally, the paper proposes a new GNN model that combines feature and label propagation with an unsupervised learning loss. The proposed methodology is evaluated on different datasets, and its performance is compared against different baseline models.

**Audience:**

Yes

**Broader Impact Concerns:**

No concerns on the ethical implications have been identified.

**Claims And Evidence:**

Yes

**Requested Changes:**

Please find the questions and requested changes in the 'Weaknesses' part of the review.

**Strengths And Weaknesses:**

Below I list the strengths and weaknesses of the paper. In the weaknesses part, I also include specific questions for the authors.

**Strengths**
- The main strength of the paper has to do with  the new datasets as well as the graph generator. Both can facilitate future research in the field.
- Besides, the paper makes interesting observations about the usage of the AUROC metric while evaluating multi-label node classification models.

**Weaknesses**
- To some extent, the main contribution of the proposed model, LFLF, is limited to an interesting combination of the feature and label propagation components. To be more specific, the feature propagation step is the same as in a GCN model (or the GraphSAGE model). The label propagation component used here is similar to traditional label propagation in semi-supervised learning.  Thus, the novel aspect of the methodology concerns the fusion by attention component.
- The description of the proposed architecture needs further clarification. First, it is not clear how the model is trained. How are the test node treated during the training phase?
- Does the model have an inductive learning capability? This point is particularly important point. Therefore, I would propose to further discuss this aspect in the paper.
- The supervised loss in Eq. (5) has been proposed before in the GraphSAGE architecture to train a GNN in an unsupervised manner. However, to my understanding, it promotes homophily-based proximity. How is this suitable for the proposed architecture?
- Looking at the experimental results in Table 3, I noticed that the GAT model outperforms GCN in two out of seven datasets. How is this explained? Besides, it seems that LFLF can be used with any message-passing architecture. Thus, I am wondering why LFLF-GAT hasn’t been considered an option here.
- The paper does not discuss the running time complexity of the model. It would be interesting to address this aspect.

---

> ### Author Response · Authors · 2023-04-30
> **Answers to Reviewer h2ys**
>
> We thank the reviewer for the suggestions. We have updated the paper with the suggestions. We marked the changes in blue to make the review easier. We welcome any further constructive discussion.
>
> **1.Novelty check**: yes , exactly our novelty comes from the fact that we combine label and feature propagation to learn an informative representation. The success of our approach is evident from our experimental results. We are a bit puzzled why simplicity and novelty of our approach is considered a weakness. We request the reviewer to kindly explain a bit more why the combination of two different techniques to fill an existing gap is a weakness.  We would also like to point out that new method is just one of the several important contributions (including a thorough analysis of the multi-label scenario for graph datasets, limitations of current methods and popularly used evaluation metric, a new graph generator model) that the current work makes.
>
> ___
> **2.Model Architecture and training**: As described in section 5 “intermediate prediction and model training”, the obtained training node embeddings from LFLF are used to train a logistic regression model to inference the labels of the test nodes. As described in section 2.1 problem setting, we follow the transductive learning setting where the features of all the nodes are available during training and the true label of the training nodes are also available during training, while the label vector of the validation and test nodes are padded using uniform padding. We use the unsupervised loss to train the model where we sample the positive neighbours from the direct neighbourhood and the negative neighbours are generated by sampling nodes from the uniform distribution.  The output of the model is the embedding of each node in the graph. We use the embeddings corresponding to the nodes in the training data to train a logistic regression classifier.
> ___
> **3.inductive learning capability**: Thanks for the comments. In the current paper, we only focus on the transductive learning setting. We did not compare the models in the inductive learning setting, but the proposed model is adaptive to the inductive learning setting. A thorough investigation of our model in inductive setting will be considered in the future work
> ___
> **4.The unsupervised loss in Eq. (5)**: We believe that the reviewer is referring to the graph proximity based loss used to train LFLF. As already explained multi-label datasets do not follow the black and white separation of low and high homophily as in the multi-class datasets. The low homophily of a multi-label dataset does not indicate the heterophily of the dataset. Although the label set of the neighbouring nodes are not completely the same, a node’s label information can still be inferred alone from the labels of its one-hop neighbours all combined together, even if it’s calculated label homophily is low. For example, in real life, we build friendship with multiple people, each of them share part of our interests.  Therefore, there is no contradiction when we optimize for proximity based loss because in fact our assumption is that a node’s neighbours are informative of its labels in a multi-label dataset even for low homophilic cases. The reason why our particular model performs better is that we learn jointly the label representations which contain the information about levels of similarity of a node with its neighbours with respect to labels.
> ___
> **5.Porformance LFLF-GAT**: Yes, our model provides plug and play scheme, where the users can also utilize other graph convolutional layers. We chose GCN layer and GraphSAGE layer as two examples of the variants of our framework. LFLF-GAT can also be used as another variant of our model.
> ___
> **6.running time complexity of the model**: Thanks for the suggestions, we added a subsection in section 5 titled “Model Variants and Space/Time Complexities.” to discuss the complexity of the model.  Please check out paper for the updates.